# TbD1 deletion as a driver of the evolutionary success of modern epidemic *Mycobacterium tuberculosis* lineages

Daria Bottai [1]*, Wafa Frigui [2], Fadel Sayes [2], Mariagrazia Di Luca [1], Dalila Spadoni[1], Alexandre Pawlik[2], Marina Zoppo[1], Mickael Orgeur[2], Varun Khanna[2,8], David Hardy[3], Sophie Mangenot[4], Valerie Barbe[4], Claudine Medigue[5], Laurence Ma[6], Christiane Bouchier[6], Arianna Tavanti [1], Gerald Larrouy-Maumus [7] & Roland Brosch [2]*

*Mycobacterium tuberculosis* (Mtb) strains are classified into different phylogenetic lineages (L), three of which (L2/L3/L4) emerged from a common progenitor after the loss of the MmpS6/MmpL6-encoding Mtb-specific deletion 1 region (TbD1). These TbD1-deleted "modern" lineages are responsible for globally-spread tuberculosis epidemics, whereas TbD1-intact "ancestral" lineages tend to be restricted to specific geographical areas, such as South India and South East Asia (L1) or East Africa (L7). By constructing and characterizing a panel of recombinant TbD1-knock-in and knock-out strains and comparison with clinical isolates, here we show that deletion of TbD1 confers to Mtb a significant increase in resistance to oxidative stress and hypoxia, which correlates with enhanced virulence in selected cellular, guinea pig and C3HeB/FeJ mouse infection models, the latter two mirroring in part the development of hypoxic granulomas in human disease progression. Our results suggest that loss of TbD1 at the origin of the L2/L3/L4 Mtb lineages was a key driver for their global epidemic spread and outstanding evolutionary success.

[1] Department of Biology, University of Pisa, Pisa, Italy. [2] Institut Pasteur, Unit for Integrated Mycobacterial Pathogenomics, CNRS UMR 3525, Paris 75015, France. [3] Institut Pasteur, Experimental Neuropathology Unit, Paris 75015, France. [4] Génomique Métabolique, Genoscope, Institut de biologie François Jacob, Commissariat à l'Energie Atomique (CEA), CNRS, Université Evry, Université Paris-Saclay, Evry, France. [5] LABGeM, Génomique Métabolique, Genoscope, Institut François Jacob, CEA, CNRS, Univ Evry, Université Paris-Saclay, 91057 Evry, France. [6] Institut Pasteur, Plate-forme génomique, Pasteur Genopole Ile de France, Paris 75015, France. [7] MRC Centre for Molecular Bacteriology and Infection, Department of Life Sciences, Faculty of Natural Sciences, Imperial College London, London SW7 2AZ, UK. [8] Present address: Hub Bioinformatique et Biostatistique, Institut Pasteur - C3BI, USR 3756 IP CNRS, Paris 75015, France. *email: daria.bottai@unipi.it; roland.brosch@pasteur.fr

Tuberculosis (TB) is a severe and complex infectious disease that remains a major cause of death in many countries[1]. Comparative genomics and whole-genome analyses have allowed reconstruction of the evolutionary pathway of its etiological agent *Mycobacterium tuberculosis* (Mtb) from a pool of recombinogenic *Mycobacterium canettii*-like strains[2–4] towards the clonal *M. tuberculosis* complex (MTBC). Within the MTBC, seven main lineages (L) of Mtb (L1, L2, L3, L4, L7) and *Mycobacterium africanum* strains (L5, L6) are known to cause TB in humans in different parts of the world[5,6]. In addition, animal-adapted MTBC strains share a common ancestor with L6 *M. africanum* strains and cause infections in different mammalian animal species[6–9]. Among the human-adapted Mtb lineages, three of them (L2/L3/L4) are particularly interesting, as they are widely spread and have diverged after a shared evolutionary bottleneck, represented by the loss of a 2153-bp genomic segment defined as Mtb-specific deletion 1 region (TbD1) (Fig. 1a)[2,10]. The TbD1-deleted (ΔTbD1) L2, L3, L4 lineages, also referred to as "modern" Mtb strains[2], of the Beijing, CAS/Dehli and Euro-American Mtb strain families[11] are often associated with globally spread TB epidemics[12–15], whereas TbD1-intact strains, also known as "ancestral" strains[2] of the East-African-Indian (EAI) strain family, rather represent endemic Mtb strains restricted to a given geographical area[15]. The TbD1-intact L1 strains are prevalent in South India and South East Asia[5], while L7 strains are restricted to the region around the Horn of Africa[16–18]. The L1 strains can be subdivided in numerous sublineages[19,20] and can cause pulmonary TB as well as extrapulmonary TB in susceptible

populations[21,22]. Several reports have suggested that Mtb strains from different lineages may induce unalike host responses[23–26].

Closer inspection of the TbD1 region showed that it encompasses the *mmpS6* and *mmpL6* genes, which in TbD1-intact strains encode members of the mycobacterial membrane protein families MmpL. In ΔTbD1 strains the *mmpS6* gene is deleted and the *mmpL6* gene truncated (Fig. 1). MmpL proteins represent a large mycobacterial protein family belonging to the RND (resistance, nodulation, and cell division) superfamily, whose members are involved in the transport of large molecules (such as various lipids and glycolipids), many of which also play a role in virulence[27–29].

The TbD1 region was first discovered by comparative genomic analyses using bacterial artificial chromosome (BAC) libraries as well as hybridization and PCR screens[2], and identified as being specifically absent from many, but not all, Mtb strains. Similar screens had also identified other large sequence polymorphisms, such as the 14 regions of difference (RD) that were absent from *Mycobacterium bovis* BCG (BCG) and present in Mtb H37Rv (RD1-RD14), or the five regions specifically absent from the Mtb H37Rv reference strain (RvD1-RvD5)[2,30]. It was thus of particular interest to investigate whether the deletion of the TbD1 region from the common ancestor of modern Mtb strains had an impact on the host-pathogen interaction and consequently on the evolutionary success of ΔTbD1 Mtb strain lineages.

Mtb strains from South India had been thoroughly studied in the 1960s by Mitchison and coworkers, whereby important differences between these isolates from Chennai (previously

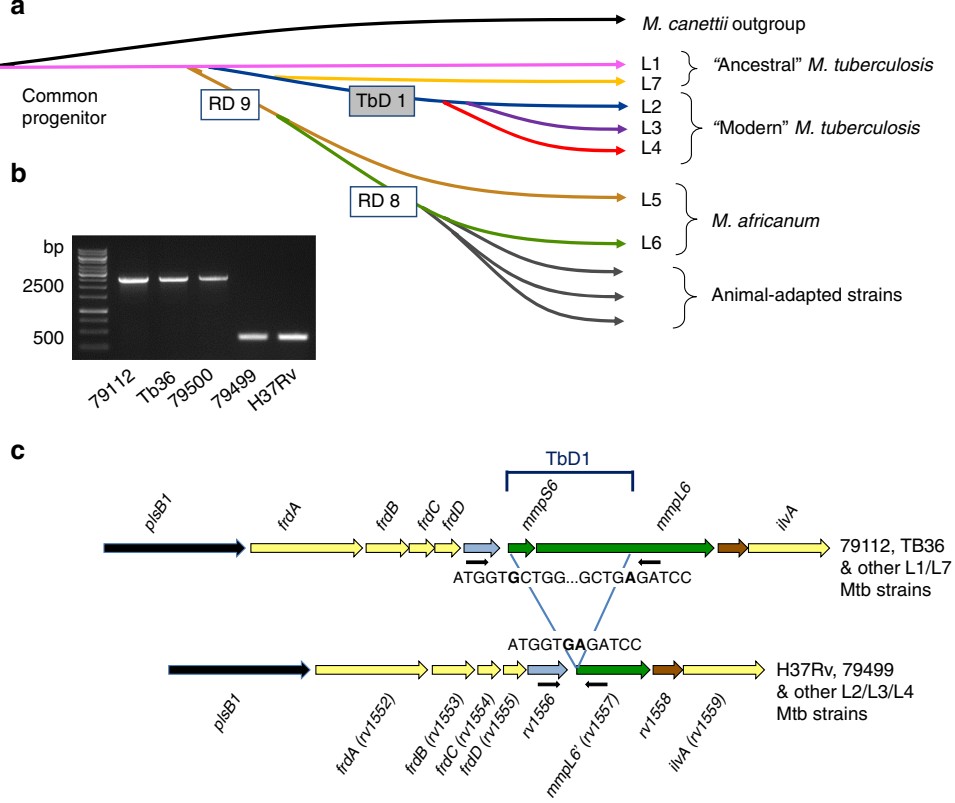

**Fig. 1 The TbD1 locus in ancestral and modern Mtb strains. a** Evolutionary scheme of the *Mycobacterium tuberculosis* complex following reference[2], showing some key large sequence polymorphisms, including the deletion of the TbD1 region, which indicates the common evolutionary origin of the lineage L2, L3 and L4 strains. **b** Amplification profiles on genomic DNAs from the different "Indian" strains, demonstrating the presence of a 2459-bp fragment (corresponding to an intact TbD1 locus) in 79112, Tb36 and 79500 strains and a 349-bp fragment resulting from the TbD1-deletion, in 79499 and H37Rv strains. Primers specific for the TbD1-flanking regions used in PCR reactions are depicted as black arrows in panel c. **c** Schematic representation of gene organization at the TbD1 locus and flanking genes in ancestral and modern Mtb strains. The sequence of the junction regions of the TbD1 locus in 79112 and H37Rv strains, as determined by genome sequencing, is also depicted.

Madras) and Mtb isolates from British TB patients were found, in terms of their virulence in the guinea pig infection model and their susceptibility to hydrogen peroxide[31]. The later independent observations that the TbD1 region was intact in most Mtb strains from Southern India[2,32] prompted us to investigate whether any link between the reported attenuated phenotype in guinea pigs of tubercle bacilli from South Indian TB patients[31] and the presence or absence of the TbD1 region might exist. We thus subjected a previously studied Indian isolate (79112)[31] that was kindly provided to us by Dr Mitchison, to whole genome sequencing and confirmed that indeed, in this strain the TbD1 region was intact, a finding which opened new opportunities for comparative studies on TbD1-intact and ΔTbD1 Mtb strains. Hence, we constructed various genetically modified Mtb strains with deleted or intact versions of the TbD1 locus (*mmpS6*/*mmpL6*) and used these mutants in numerous in vivo and in vitro assays in comparison with their corresponding TbD1-intact and ΔTbD1 wild-type (WT) Mtb strains and Mtb strains from different phylogenetic lineages. Our findings strongly suggest that the loss of the TbD1 region was a key event in the evolutionary success of the modern L2, L3 and L4 Mtb lineages.

## Results

**Genome analysis of Mtb strains.** Selected Mtb strains were subjected to whole-genome sequencing and/or PCR analysis using primers specific for the 5'- and 3'-TbD1 genomic flanking regions to determine whether or not the TbD1 region was present in the respective strains. Among these strains, some (79112, 79499, 79500) were originally isolated by the team of Dr Mitchison in the 1960s in South India and referred to as Indian Mtb strains. These strains showed different virulence profiles after intramuscular injection in guinea pigs[31]. Moreover, we also included the Mtb strain Tb36 from the collection of Dr van Soolingen[14]. Mtb Tb36 is a patient isolate of Indian origin and was identified as one of the first TbD1-intact Mtb strains in our previous study on the evolutionary pathway of the tubercle bacilli[2]. Combined data from genome and PCR analyses revealed that strains 79112, Tb36 and 79500 were TbD1-intact L1 strains harboring the 2153-bp sized intact TbD1 genomic locus, while strain 79499 represented a modern ΔTbD1 L4 strain (Fig. 1). Apart from determining the presence or absence of the TbD1 region, the generated genome data of selected TbD1-intact Mtb strains also served for determination of the phylogenetic lineages and sequence comparisons of selected genes involved in global mycobacterial regulation processes.

**Virulence evaluation of Mtb in cellular and animal models.** Given the reduced virulence observed in the 1960s for the Mtb 79112 strain in guinea pigs[31,33], we first sought to validate our guinea pig aerosol infection model by including some of these previously used Mtb strains into our infection experiments. We thus generated and analyzed the virulence profiles of TbD1-intact strains (79112, 79500 and Tb36) (Fig. 1b) and compared them with those of ΔTbD1 79499 clinical and H37Rv reference strains (Fig. 1b). This analysis revealed that strain 79112 indeed showed a strongly reduced virulence phenotype relative to H37Rv and 79499 strains in guinea pigs, and importantly, lower virulence was also seen for the two other TbD1-intact strains in this model (Fig. 2a–c). A ~2-log lower CFU level was recovered from lungs of Mtb 79112- and Tb36-infected animal as compared to that detected in guinea pigs infected with 79499 and H37Rv Mtb strains (Fig. 2a, c and Supplementary Fig. 1). Similarly, a 1-log reduction was observed in CFU values recovered from spleens of 79112- and Tb36-infected guinea-pigs relative to 79499- and H37Rv- infected animals (Fig. 2b). Mtb 79500 also showed lower

CFU values in lungs and spleens than TbD1-deleted strains, although the effect was less pronounced (Fig. 2a–c), a finding which is in agreement with previous observations[31]. We thus focused on strains 79112- and Tb36 for further experiments.

Interestingly, the evaluation of the virulence profiles of Mtb 79112 and Tb36 in the C57BL/6 standard murine aerosol infection model revealed no significant differences in the replicative potential of these TbD1-intact strains relative to the H37Rv control (Fig. 2d–f).

Finally, we also evaluated the intracellular growth of the strains in different ex vivo cellular models, and found that TbD1-intact strains again displayed similar growth kinetics as TbD1-deleted strains in murine bone marrow-derived macrophages (Supplementary Fig. 2a, b), whereas they showed reduced intracellular growth in PMA-activated THP-1 human macrophage-like cells (Supplementary Fig. 2 e, f) and in A549 Type II pulmonary epithelial cells (Supplementary Fig. 2g, h). Such reduced growth of TbD1-intact ancestral *M. tuberculosis* strains relative to ΔTbD1 modern strains in THP-1 cells agrees with recent observations obtained after infections of donor-derived human macrophages with different Mtb strains, where ancestral strains also showed much lower intracellular replication than strains of modern TbD1-deleted lineages[26]. Taken together, our data revealed significantly decreased virulence of TbD1-intact 79112 and Tb36 Mtb strains compared to strains of modern lineages in the guinea pig aerosol infection model, as well as the relative attenuation of the TbD1-intact strains in cellular infection models involving human phagocytes. In contrast, no significant in vivo growth differences between TbD1-intact and ΔTbD1 Mtb strains were observed in the standard C57BL/6 murine model, in agreement with similar results from previous infection experiments involving Mtb H37Rv and TbD1-intact Mtb strains in Balb/C[34] and C57BL/6 mice[3], which lasted up to 8 weeks[34] and 13 weeks[3], respectively. These results emphasize the potential impact of the choice of the infection model on the evaluation of mycobacterial virulence, which may differ in the physiological and ecological conditions and may produce diverging predictions for human global TB epidemiology.

**Construction of TbD1 knock-out and knock-in Mtb strains.** To investigate more in depth the link between an intact TbD1 locus and the attenuation profile in guinea pigs we used two complementary genetic approaches, which consisted in the deletion of the TbD1 region from strain 79112, and in the complementation of strain H37Rv with a functional TbD1 locus from strain Tb36. The 79112ΔTbD1 mutant, in which the TbD1 locus is replaced by a kanamycin resistance gene, was generated by using the recombineering method[35] (Fig. 3a). PCR amplification profiles on genomic DNAs from 79112 WT and 79112ΔTbD1 strains confirmed the replacement of TbD1 with the kanamycin resistance gene in the mutant. For complementation purposes, we constructed integrative cosmid 2G12, by cloning the TbD1 locus and flanking regions from Mtb strain Tb36 into the pYUB412 cosmid vector backbone[36], which allowed potential operon structures within the genomic environment of the *mmpS6*/*mmpL6* locus to be conserved (Supplementary Fig. 3). The 79112ΔTbD1-C complemented control strain was obtained by integration of the 2G12 cosmid into the *attB* site of the genome of the 79112ΔTbD1 mutant (Fig. 3a, and Supplementary Fig. 3). The integration of a functional TbD1 locus in this strain was confirmed by PCR (Fig. 3b). The same approach was used for the complementation of H37Rv with a full-sized TbD1 locus (Fig. 3a). The correct integration of the TbD1 locus in two different H37Rv::TbD1 clones (clones 1 and 2) was tested by PCR analysis using primers amplifying the entire *mmpS6*/*mmpL6* locus (Fig. 3c), and the

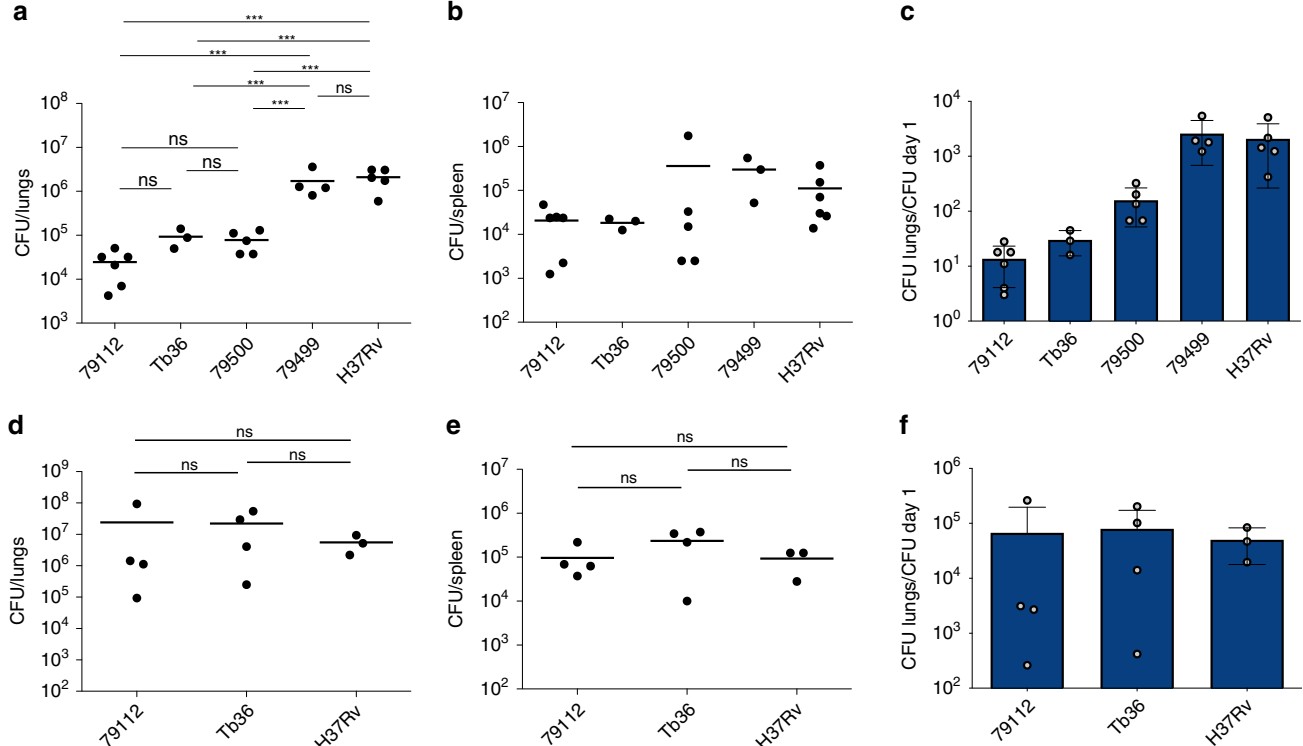

**Fig. 2 Virulence profiles of selected Indian Mtb WT strains in different in vivo models. a–c** In vivo growth profiles of TbD1-intact and ΔTbD1 Mtb strains in the guinea pig infection model. Guinea pigs were aerosol-challenged with different Indian strains or the laboratory reference strain H37Rv. Eight weeks after infection, the bacterial load in lungs (**a**) and spleen (**b**) was determined. Statistical differences in CFU values obtained in organs from animals infected with different Mtb strains were determined by the univariate analysis of variance test followed by Sidak post-hoc test (***$P < 0.001$; ns: not significant). **c** CFU ratio (CFU at day 56/CFU at day 1) obtained in lungs of animals infected with Mtb Indian strains and the H37Rv control strain. **d–f**. Virulence potential of ancestral Mtb strains (79112, Tb36) in the mouse model in comparison with H37Rv. Groups of four C57BL/6 mice were aerosol infected with different Mtb strains, to obtain an infection dose of 100–500 CFU/lungs. Four weeks after infection, the bacterial load in lungs (**d**) and spleen (**e**) of Mtb-infected animals was determined. Statistical significance of differences in CFU numbers was determined by one-way Anova test followed by Bonferroni post hoc test (ns: not significant). **f** CFU ratio (CFU at day 30/CFU at day 1) obtained in lungs of animals infected with TbD1-positive strains and the H37Rv control strain. The figure depicts single data points, mean, and standard deviation of CFU numbers or CFU ratio values obtained in representative guinea pig (**a–c**) and mouse (**d–f**) infection experiments.

constructs are named hereafter Mtb H37Rv::TbD1. In an assessment of their in vitro growth abilities, the various WT, mutant and complemented strains were grown in different liquid growth media (e.g., Middlebook 7H9 and Dubos media) at different temperatures (37 and 39 °C), where they displayed comparable growth kinetics (Supplementary Fig. 4). In contrast, TbD1-intact Mtb 79112 and Tb36 strains displayed smaller colony morphotypes on solid media compared to the ΔTbD1 H37Rv strain (Supplementary Fig. 5).

We next evaluated the potential effect of the presence or absence of the TbD1 region on intracellular growth kinetics of the different mutant strains in two different cellular models, namely the PMA-activated THP-1 human macrophage and in A549 Type II pulmonary epithelial cell lines. No differences were detected among TbD1-intact and ΔTbD1 strains in the percentage of uptake by THP-1 or A549 cells, at the multiplicity of infection tested (m.o.i. 20:1 and 1:1 cell:bacteria for THP-1; m.o.i. 10:1 and 1:1 cell:bacteria for A549 cells) (Fig. 4a, d). However, increased intracellular growth was observed over a 6-day period for the TbD1-deleted mutant 79112ΔTbD1 as compared to the corresponding WT and complemented strains (79112 and 79112ΔTbD1-C), both in THP-1 (m.o.i 20:1) and A549 (m.o.i 10:1) cells (Fig. 4b, c, e, f). Similarly, the complementation of H37Rv with a functional TbD1 locus resulted in a significative reduction of the intracellular growth ability in both cellular models analyzed (Fig. 4b, c, e, f).

**Virulence of TbD1 mutants in guinea pigs and C57BL/6 mice**. The virulence profiles of TbD1-deleted and complemented mutant strains were then evaluated in aerosol-infected guinea pigs, and compared to the corresponding 79112 and H37Rv WT control strains. Data obtained in the Mtb 79112 background revealed that the deletion of the TbD1 locus conferred an improvement of the in vivo replicative potential, resulting in a 2 log and 1 log increase in CFU values recovered from lungs and spleen, respectively, of animals infected with the mutant strain relative to that obtained in guinea pigs infected with the WT strain (Fig. 5a–c; Supplementary Fig. 6a). The complementation of the 79112ΔTbD1 mutant restored the in vivo attenuation profile (Fig. 5a–c). Consistently, the integration of an intact TbD1 locus into the H37Rv genome resulted in a significant attenuation of the corresponding recombinant strain in the guinea pig infection model. A 1.5 log and 2 log reduction was indeed observed in CFU values obtained from lungs and spleens of H37Rv::TbD1-infected animals as compared to H37Rv WT-infected guinea pigs (Fig. 5d–f; Supplementary Fig. 6b).

In contrast, when tested in the C57BL/6 mouse infection model (Supplementary Fig. 7), like seen for different WT strains described above, genetically engineered TbD1-intact and ΔTbD1 variants showed similar virulence profiles. Comparable CFU values were observed in lungs and spleens of mice, aerosol-infected with the 79112ΔTbD1 mutant or its corresponding 79112 WT strain (Supplementary Fig. 7a–d), similar to the

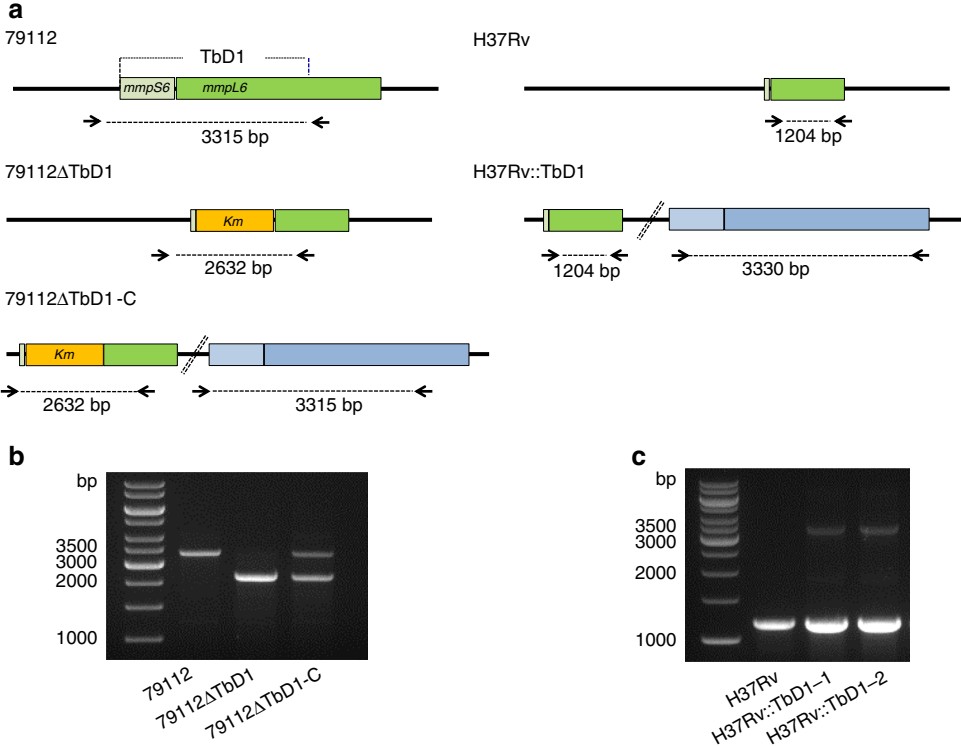

**Fig. 3 Deletion and integration of the TbD1 locus in the 79119 and H37Rv genetic backgrounds, respectively. a** Schematic representation of the genomic organization of the TbD1 locus in the 79112ΔTbD1 mutant and complemented strain (79112ΔTbD1-C). The schematic representation of the genomic organization of recombinant H37Rv strains, harboring an intact TbD1 locus is also depicted. Arrows indicate primers used in PCR reactions. **b**, **c** Amplification profiles obtained in PCR reactions performed on genomic DNAs from different Mtb strains by using primer pairs, specific for TbD1 flanking region. The 2632 bp-fragment obtained in 79112ΔTbD1, or the 2632-bp and 3315-bp amplification products detected in 79112ΔTbD1-C confirmed the TbD1 deletion and its replacement by a kanamycin resistance gene in the 79112ΔTbD1 mutant, the correct TbD1 deletion/re-integration in the 79112ΔTbD1-C complemented derivative strain, respectively (**b**). Similarly, the 1204-bp product obtained in H37Rv, or the 1204-bp and 3330-bp fragments observed in H37Rv::TbD1 clones correspond to the TbD1-deleted locus originally present in WT H37Rv and to the full-size TbD1 region (*mmpS6/mmpL6*) integrated into the genome, respectively (**c**).

situation of Mtb H37Rv and its derivative H37Rv::TbD1 (Supplementary Fig. 7a–f).

**TbD1 deletion enhances Mtb virulence in C3HeB/FeJ mice**. To get deeper insights into potential factors that might have contributed to the divergent results obtained for virulence comparisons of TbD1-intact and ΔTbD1strains in guinea pigs and C57BL/6 mice, we selected Mtb strains 79112, 79112ΔTbD1, 79112ΔTbD1-C and H37Rv, to evaluate their virulence in the C3HeB/FeJ mouse model, known for developing hypoxic necrotic lung granulomas 8–14 weeks after low-dose infection with Mtb[37–39]. Hence, C3HeB/FeJ were infected with different strains (10–20 CFU/lungs), and the bacterial load in selected organs was determined 4, 10 and 14 weeks after infection, which correspond to the acute (4 weeks) and chronic (10-14 weeks) phase of infection[38]. Comparable CFU numbers were recovered from mice infected with TbD1-intact strains (79112 and 79112ΔTbD1-C*)* and ΔTbD1 strains (79112ΔTbD1 and H37Rv) at 4 weeks of infection (Fig. 6a, b and Supplementary Fig. 8a), confirming previous observations in the standard C57BL/6 mouse model. However, a significant lower bacterial load was observed for strain 79112 in the lungs and spleens of these mice in comparison with 79112ΔTbD1 and H37Rv strains after 14 weeks of infection (Fig. 6 c, d), confirming a trend observed already when bacterial counts were determined in organs at 10 weeks post-infection (Supplementary Fig. 8b). At these long-term time points post-infection, mice infected with strain 79112ΔTbD1-C showed an overall reduced level in bacterial

load in lungs and spleens at 10 and 14 weeks post-infection. Inspection of histopathological sections of the lungs showed that all of the tested strains induced granuloma formation in this model (Supplementary Fig. 9), known for some heterogeneity among individual mice[40].

**Search for TbD1-associated lipid substrates**. Given the reported implication of several MmpS/MmpL systems in mycobacterial lipid transport[29,41], we evaluated whether the TbD1-encoded MmpS6/MmpL6 proteins might be involved in lipid transport, and subjected a set of WT and mutant Mtb strains to selected lipidomics assays, for which the strains were cultured under aerobic or anaerobic conditions. Using different solvent conditions and thin-layer chromatography (TLC), as well as MALDI-ToF mass spectrometry (MS) in the negative ion mode, lipid preparations were examined. This methodology allows a snapshot of key membrane phospholipids such as phosphatidyl-myo-inositol mannosides (PIMs) and potential lipid virulence factors, e.g. (sulfolipid, SL-I) to be taken[42–44].

As shown in Supplementary Fig. 10, TLC of extractible lipids from four different WT and mutant strains in various solvent mixtures produced comparable profiles, where no clear differences between strains were noticed. Moreover, MS analysis of a second series of sample preparations of 5 strains, displayed a set of peaks at m/z 851.6 assigned to phosphatidyl-myo-inositol (PI), *m/z* 1175.7, *m/z* 1413.9 and *m/z* 1694.1 assigned to phosphatidyl-myo-inositol dimannoside with 2 (PIM$_2$), 3 (Ac$_1$PIM$_2$) and 4 (Ac$_2$PIM$_2$) fatty acid residues, *m/z* 2061.1

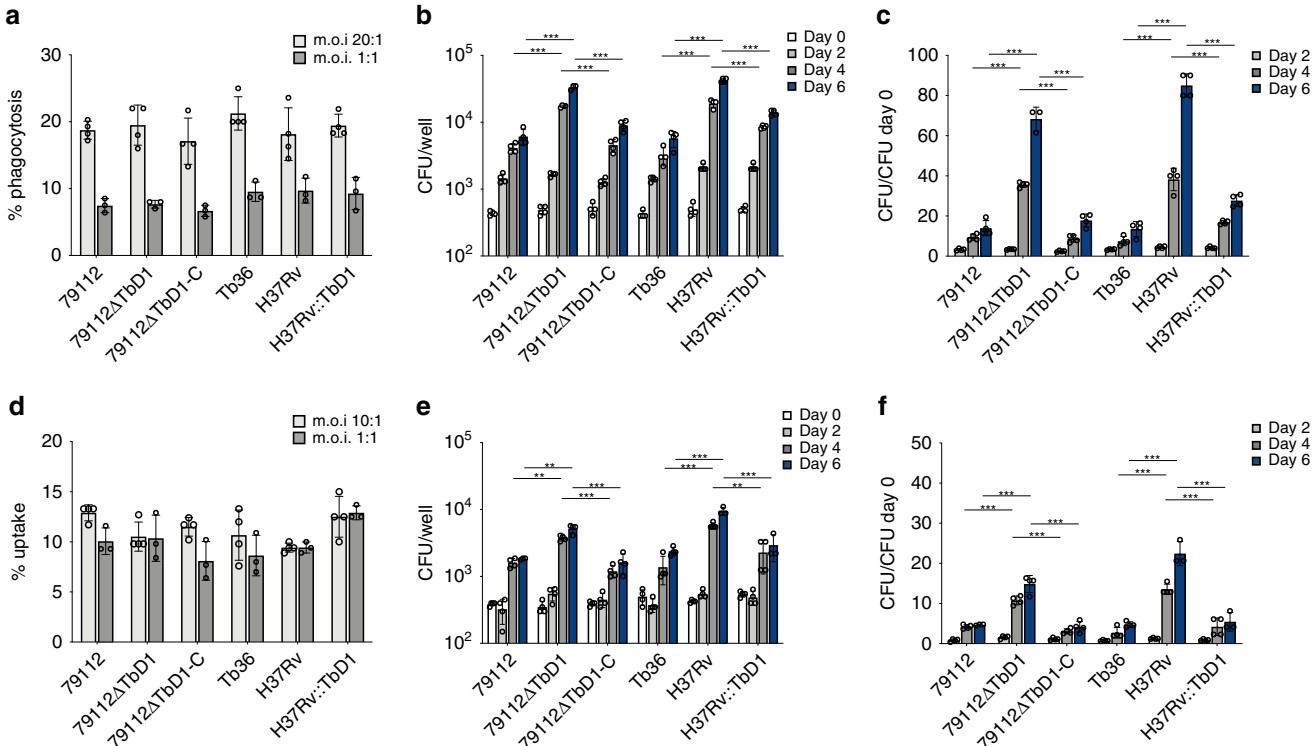

**Fig. 4 Intracellular growth profiles of TbD1 WT and mutant strains in THP-1 and A549 human cell lines. a–c** Comparative analysis of the percentage of phagocytosis (**a**) and growth (**b**, **c**) of TbD1-intact and ΔTbD1 WT and mutant strains in human THP-1 cells. **d–f** Comparative analysis of the percentage of uptake (**d**) and growth (**e–f**) of TbD1-intact and ΔTbD1 WT and mutant strains in human A549 cells. For determination of the percentage of uptake, THP-1 cells were infected at m.o.i 20:1 and 1:1 (cells: bacteria), while A549 cells were infected at m.o.i 10:1 and 1:1 (cells: bacteria). The comparison of growth kinetics of different TbD1-intact and ΔTbD1 WT and complemented strains was undertaken at m.o.i 20:1 (cells:bacteria) for THP-1, and 10:1 (cells:bacteria) for A549. These m.o.i ensured the integrity of the infected cell monolayer over a 6-day period. In both these ex vivo models, the numbers of intracellular bacteria (CFU) were determined immediately after phagocytosis and at different time points (as indicated in panels **b**, **e**). CFU ratio values (CFU at different time points/CFU at day 0) are reported in **c**, **f**. The statistical significance of differences in CFU and CFU ratio values among the strains were determined by one-way Anova with Bonferroni post hoc test. Only the statistical significance of differences between the 79112ΔTbD1 mutant and its corresponding control strains (79112 and 79112ΔTbD1-C) or H37Rv and its derivative H37Rv::TbD1 and related control Tb36, at days 4 and 6 post infection are depicted in the figure (**P < 0.01; ***P < 0.001). The figure shows the single data points, mean and standard deviations of uptake percentage, CFU numbers and CFU ratio values obtained in a representative experiment performed in quadruplicate (or in triplicate for determination of uptake percentage at m.o.i 1:1).

assigned to phosphatidyl-myo-inositol hexamannoside containing 3 fatty acids. The high mass range displays a set from $m/z$ 2100 up to $m/z$ 2800 assigned to SL-I[44]. Similar to TLC analyses, this analysis did not identify notable differences between strains. (Supplementary Fig. 11). We finally also analyzed the mass spectra of mycolic acid methyl esters (MAMEs) from a set of WT and mutant strains, and again found no eminent differences (Supplementary Fig. 12). Hence, our initial screening of selected phospholipid profiles and MAMEs did not identify potential TbD1-associated lipid factors and suggests that dedicated fine structure analyses and/or analyses under different growth conditions will be necessary to gain deeper insights into the issue.

**Identification of TbD1-associated stress sensitivity factors.** In order to find possible explanations for the virulence differences observed between TbD1-intact and ΔTbD1 strains in the guinea pig and the C3HeB/FeJ models, selected stress response profiles of WT and mutant Mtb strains were tested. As the 79112 and other Indian Mtb strains were described in the 1960s by Mitchison and colleagues as being more susceptible to $H_2O_2$ than British strains[31], we first incubated the bacterial cultures of different Mtb lineages in the presence of 10 mM $H_2O_2$ for 1 h at 37 °C, and determined the CFU. By this assay, in agreement with initial studies of Mitchison and coworkers[31], the TbD1-intact 79112 and

Tb36 strains both displayed an increased susceptibility to $H_2O_2$ in comparison with ΔTbD1 strain types, as shown by their strongly reduced survival percentage relative to Mtb H37Rv (Fig. 7a). We then further corroborated the observed lineage-specific sensitivity to oxidative stress by testing more clinical isolates from different lineages (Fig. 7b, c). Notably, the EAI 1400-10134 and EAI 1400-10144 L1 strains, which were confirmed by PCR analyses as TbD1-intact strains (Fig. 7b) revealed an increased sensitivity to $H_2O_2$ as compared to H37Rv and other TbD1-deleted strains of lineages L2 and L3 (Fig. 7c). The survival percentages observed ranged from 0.17 to 1.83% for the TbD1-intact strains, and from 9.62 to 17.5% for Mtb H37Rv and L3 strains Delhi 2004, Delhi 2005-0318, Delhi 2004-1592. Finally, the three tested L2 Beijing strains displayed a hyper-resistant phenotype to $H_2O_2$ (survival percentage 60%), a finding which might be caused by additional, Beijing strain-specific genetic determinants[45].

The increased susceptibility of ancestral Mtb strains to reactive oxygen intermediates (ROI) was further confirmed by comparisons of the diameters of the inhibition growth zones obtained for 79112 and Tb36 strains as compared to Mtb H37Rv after exposure to 350 nmol cumene hydroperoxide (an organic peroxide) or 50 nmol plumbagine (a superoxide generator) in disk diffusion assays on solid medium (Supplementary Fig. 13a, b).

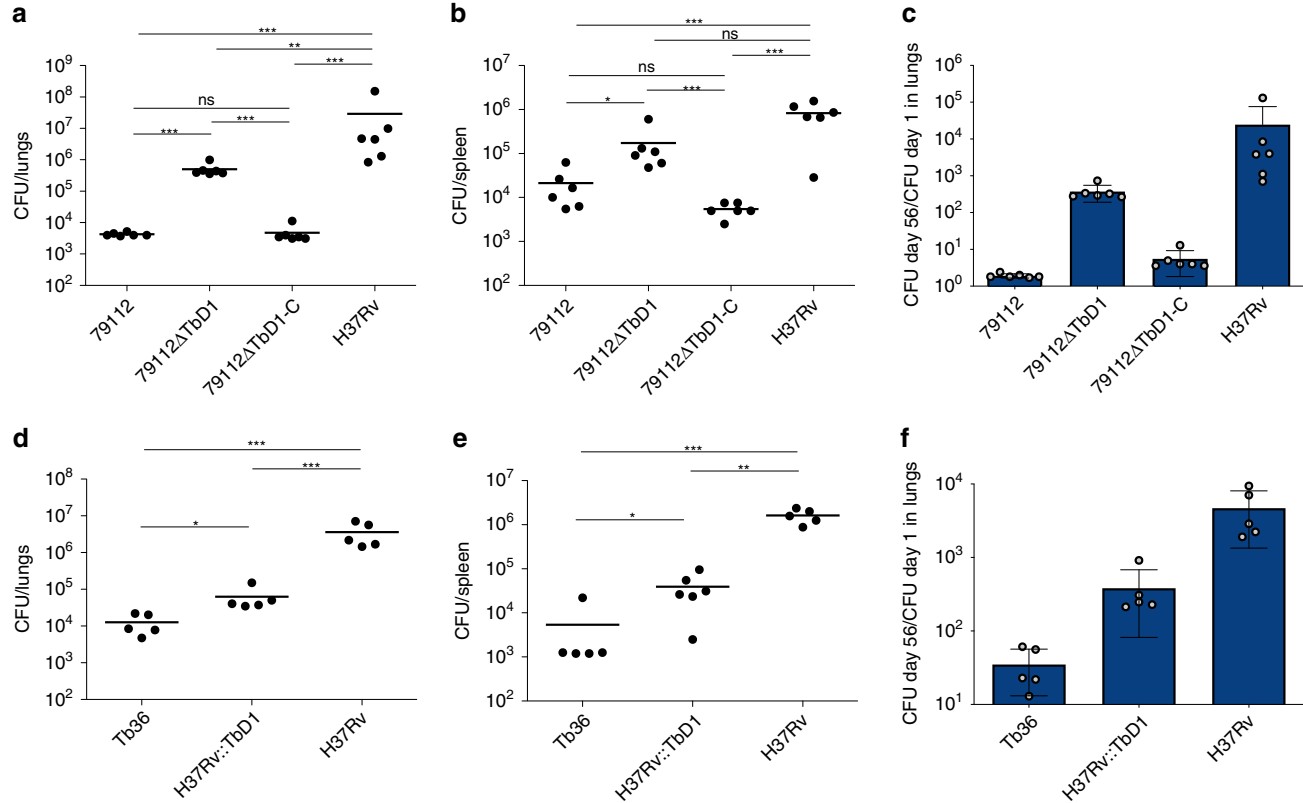

**Fig. 5 Impact of TbD1 on Mtb virulence in guinea-pigs.** Guinea pigs were aerosol infected with a panel of TbD1-deleted or TbD1-complemented Mtb modified strains. Eight weeks after challenge, the bacterial load in target organs was determined. **a**, **b** CFU numbers obtained in lungs (**a**) and spleen (**b**) of guinea-pigs infected with 79112ΔTbD1 or 79112ΔTbD1-complemented strains. Animal control groups were infected with 79112 WT or H37Rv strains. **c** CFU ratio (CFU at day 56/CFU at day 1) obtained in lungs of animals infected with 79112-derivative strains and 79112 or H37Rv WT strains. **d**, **e** CFU numbers obtained in lungs (**d**) and spleens (**e**) of guinea-pigs infected with the H37Rv::TbD1 strain, obtained by complementation of strain H37Rv with the TbD1 locus from strain Tb36 via cosmid 2G12. As controls, guinea pigs were infected with Tb36 and H37Rv parental strains. **f** CFU ratios (CFU at day 56/CFU at day 1) obtained in lungs of animals infected with the H37Rv::TbD1 strain, complemented with the TbD1 region from Tb36 via cosmid 2G12. Statistical differences in CFU values obtained in organs form animals infected with different Mtb strains were determined by the univariate analysis of variance test followed by Sidak post-hoc test (*$P < 0.05$; **$P < 0.01$; ***$P < 0.001$; ns: not significant). The figure reports single data points, mean and standard deviation of CFU number or CFU ratio values obtained in representative experiments performed with 6 (**a–c**) or 5 (**e–f**) animals per group.

Most importantly, a significant increase in resistance to 10 mM $H_2O_2$ was observed for the genetically defined 79112ΔTbD1 deletion mutant relative to corresponding 79112 WT and complemented 79112ΔTbD1-C strains (Fig. 7a). In contrast, reduced resistance to 10 mM $H_2O_2$ was noted for two independent TbD1-complemented H37Rv::TbD1 knock-in clones (Fig. 7a). Our results thus show a correlation between the presence of an intact TbD1 locus and increased susceptibility of Mtb to ROI.

To investigate whether the presence of an intact TbD1 region also resulted in an increased sensitivity to other stress conditions that play a role during infection in an intracellular environment, the susceptibility of two selected TbD1-intact Mtb strains (79112 and Tb36) to reactive nitrogen intermediates (RNI) and/or acidic pH was tested in comparison with the H37Rv control. Strains were grown in Middlebrook 7H9 medium, diluted in fresh medium and incubated at 37 °C in presence of 10 mM sodium nitroprusside (SNP). Control cultures in fresh Middlebrook 7H9 medium were also included. After 1 and 4 days of incubation, the CFU numbers were determined and the survival percentage relative to the day 0 was calculated. As reported in Supplementary Fig. 13c, the exposure to 10 mM SNP resulted in a progressive reduction of the survival percentage of all tested strains and no relevant differences in survival percentages were observed

between ancestral and H37Rv strains. Similar results were obtained when 79112, Tb36 and H37Rv strains were exposed to the nitrosative stress generated by $NaNO_2$. Bacterial cultures were obtained in Middlebrook 7H9 medium as reported above, diluted in fresh Middlebrook 7H9 medium acidified to pH 5.5, and incubated in presence of $NaNO_2$ at two different concentrations (1 and 10 mM) for 1 and 4 days. The assay was performed in Middlebrook 7H9 medium at pH 5.5 as $NaNO_2$ is active at acidic pH. Control cultures in Middlebrook 7H9 pH 7.2 were also included. As shown in Supplementary Fig. 13c, no statistically significant differences in survival percentages for TbD1-intact strains and H37Rv were found neither after incubation at pH 5.5 alone, nor after exposure to $NaNO_2$.

**TbD1 locus impacts Mtb survival under hypoxic conditions.** Hypoxia is one of the main stresses characterizing the intracellular environment during the latency phase of Mtb infection. Under these conditions, Mtb is able to survive, although in a non-replicative and metabolically non-active state, which has been described as "dormancy state"[46]. In order to compare the ability of ancestral and modern Mtb strains to survive in hypoxic conditions and investigate the potential impact of the TbD1 locus on this important feature, the growth kinetics/survival of Mtb TbD1-intact and TbD1-deleted strains were compared in the Wayne

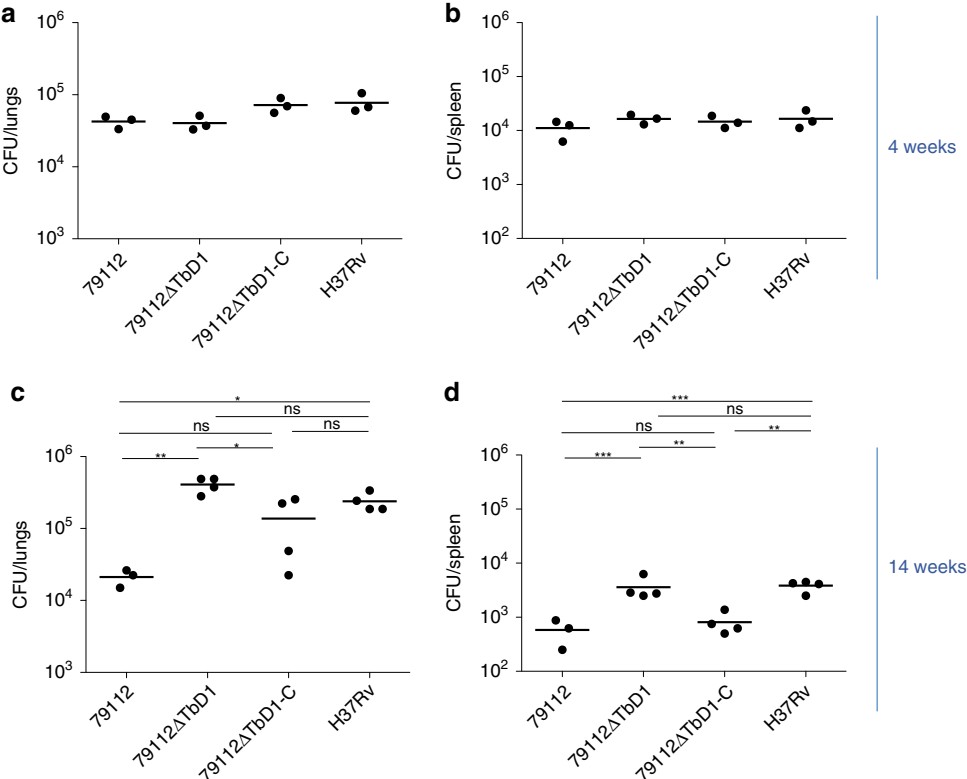

**Fig. 6 Growth kinetics of 79112 and isogenic ΔTbD1 and TbD1-complemented strains in C3HeB/FeJ mice.** C3HeB/FeJ mice were aerosol infected with the Mtb 79112ΔTbD1 mutant and its corresponding TbD1-intact 79112 parental and 79112ΔTbD1-C complemented strains (10–15 CFU/lungs). Mtb H37Rv was included as an additional control strain. Four (**a**, **b**) and 14 (**c**, **d**) weeks after infection, the bacterial loads in lungs (**a**, **b**) and spleens (**c**, **d**) were determined. Statistical differences in CFU values obtained in organs from mice infected with different Mtb strains were determined by one-way Anova with Bonferroni post hoc test (*$P < 0.05$; **$P < 0.01$; ns: not significant). The figure reports single data points, mean and standard deviation of CFU number or CFU ratio values obtained in a representative experiment performed with 3 (**a**, **b**) or 4 (**c**, **d**) mice per group.

model of progressive oxygen depletion, in vitro. Bacteria were grown in Dubos medium without glycerol until cultures reached an $OD_{600} = 0.4$, and then inoculated in Dubos fresh medium in 30 ml sealed tubes, in order to obtain a liquid volume: air volume ratio of 2:1. After 2, 5, 10, 25, and 40 days of incubation at 37 °C the number of viable bacteria was determined. As reported in Fig. 7d, no significant differences were observed among the strains in CFU values obtained after 2 and 5 days of incubation, corresponding to the active replication phase (day 2) or to the non-replicative state 1 (day 5). Starting from the day 10 (non-replicative state 2) the CFU number of the Mtb H37Rv WT strain remained constant, while the CFU values of strains 79112 and Tb36 decreased progressively. These data indicate a reduced survival ability of TbD1-intact strains under hypoxic conditions compared to the TbD1-deleted H37Rv control strain.

In the same model, the genetic mutants 79112ΔTbD1 and 79112ΔTbD1-C, as well as the H37Rv::TbD1 strain displayed similar growth kinetics until day 10 of incubation. In contrast, starting from day 10, the survival of strains strictly correlated with the absence/presence of the TbD1 locus. While the CFU values for 79112ΔTbD1 remained constant and comparable to those obtained for the H37Rv WT until day 40, the CFU values of TbD1-intact H37Rv::TbD1 and 79112ΔTbD1-C strains progressively reduced (Fig. 7d). Similarly to the Tb36 and 79112 WT strains, a 1-log reduction was observed in CFU values recovered for 79112ΔTbD1-C and H37Rv::TbD1 strains as compared to their corresponding parental strains (Fig. 7d). These data support a direct role of TbD1-encoded gene products in sensitivity to hypoxic conditions.

## Discussion
One of the key questions in the overall TB epidemiology is why and how certain clones of Mtb strains have evolved into globally distributed, epidemic strains and strain families, whereas others have remained primarily endemic strains restricted to certain geographic areas and their local human populations. First hints for a molecular marker that could distinguish between predominantly epidemic and endemic Mtb strains were obtained by comparative analyses of the regions of difference in genomes of selected MTBC members[2]. The thereby identified Mtb-specific deletion region TbD1 was deleted in almost all Mtb strains from a large strain collection used in an inter-laboratory molecular epidemiological marker study[2,14], whereas the other MTBC members (*M. africanum, M. microti, M. bovis*) all harbored an intact TbD1 region. The few Mtb strains in which the TbD1 region was found intact, had few IS*6110* elements and were of East-African or Indian origin[2]. These strains showed a particular spoligo-type signature that was later confirmed to be characteristic for "ancestral", TbD1-intact Mtb strains from South India and South-East Asia, the so-called East-African-Indian (EAI) strains[32,47,48]. Whole-genome-based phylogeny studies have later defined these TbD1-intact strains as lineage 1 (L1) strains, prevalent in East Africa, the Philippines and the Rim of Indian Ocean[5]. In these TB endemic regions, TbD1-intact strains may represent more than a third of all patient isolates[11,32,48], whereas in most other geographic regions with high TB burden, the vast majority of Mtb patient isolates corresponds to ΔTbD1 strains of the L2, L3 and L4 lineages, also known as Beijing (L2), CAS/Dehli (L3), Haarlem/Ural/X/LAM (L4) strain families according

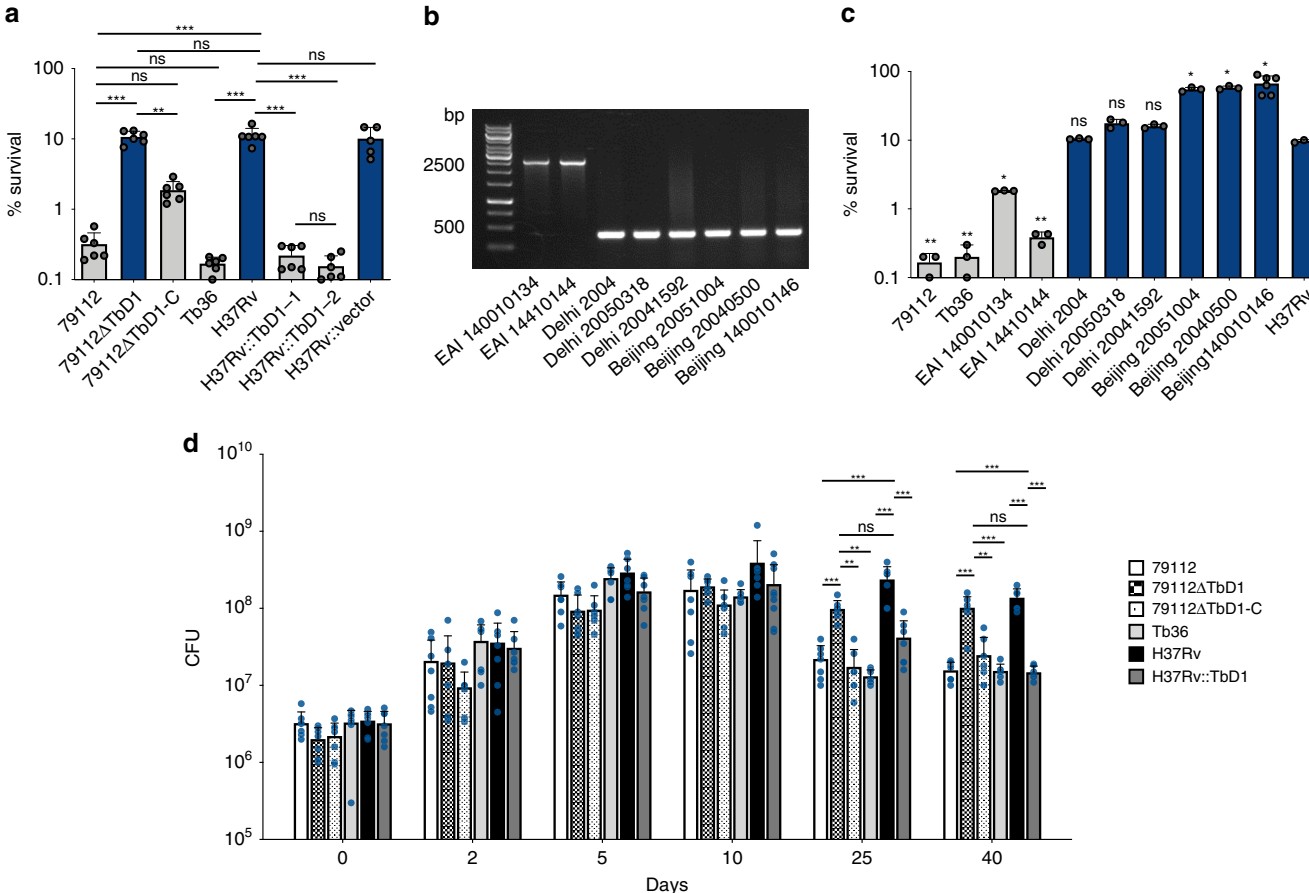

**Fig. 7 Impact of TbD1 on Mtb sensitivity to oxidative stress and hypoxia. a** Survival percentages of WT 79112 and Tb36 ancestral TbD1-intact strains or modern Mtb H37Rv, in comparison with those of TbD1-deleted or TbD1-complemented derivative strains after 1-h exposure to 10 mM $H_2O_2$. An H37Rv strain complemented with the empty vector was included in the assay as an additional control. The susceptibility of each strain was expressed as percentage of survival relative to the time 0. The figure depicts single data points, mean and standard deviation of survival percentages obtained for each strain in six independent bacterial cultures. **b**, **c** Sensitivity profiles of a panel of Mtb clinical isolates belonging to different lineages, harboring or not an intact TbD1 locus. **b** Amplicons obtained by PCR analysis on genomic DNAs from different Mtb isolates, by using primers specific for the TbD1-flanking regions. **c** Survival percentages of clinical isolates after exposure to $H_2O_2$. Mtb 79112, Tb36 and H37Rv were also included as control strains. In **a**, **c** TbD1-intact strains are indicated by gray bars, while TbD1-deleted strains are indicated by blue bars. The graph represents single data points, mean and standard deviation of survival percentages obtained for each strain in a representative experiment performed in triplicate (6 bacterial replicates were testes for the Beijing 140010146 strain). **d** Growth and survival of Mtb TbD1-intact and TbD1-deleted strains in the Wayne dormancy culture system. All strains were grown in Dubos medium without glycerol, in sealed 30 ml tubes (liquid volume: air volume ratio of 2:1). After 2, 5, 10, 25, and 40 days of incubation at 37 °C the CFU numbers were determined. The panel shows the single data points, mean and standard deviation of CFU values recovered for each strain in seven independent growth/survival assays. Statistical significance of differences in survival percentages (**a**, **c**) or in CFU values (**d**) determined by one-way Anova with Bonferroni post hoc test (*$P < 0.05$; **$P < 0.01$; ***$P < 0.001$; ns: not significant) are shown. In **c** only the statistical significance of differences in survival percentages between different clinical isolates and the H37Rv reference strains is depicted.

to the spoligotyping nomenclature[12,14,15,49–52]. These latter lineages are thought to have evolved from a common ancestor that underwent deletion of the TbD1 region, and they represent the great majority of globally-spreading Mtb strains, showing an enhanced potential to out-compete and replace local endemic Mtb strains[53,54]. Given these particular characteristics of the global TB epidemiology, it was our aim to evaluate whether the loss of the TbD1 region might account for yet undetected fitness advantages of L2, L3, and L4 strains that are linked to their enhanced global spreading and epidemic character. We were particularly interested in this subject, as a series of early reports dating from the pre-genomic era had suggested that cultures of tubercle bacilli isolated from South Indian tuberculosis patients differed from corresponding cultures from British patients in their virulence in the guinea pig and their susceptibility to the bactericidal action of hydrogen peroxide[31,55].

In the present study, by the use of an integrated pathogenomic approach, involving whole-genome sequencing and construction of knock-in/knock-out mutants of Mtb strains from different phylogenetic lineages, as well as virulence studies in guinea pig, mouse and cellular infection models, we were now able to attribute these previous observations to the loss of the TbD1 region from L2/L3/L4 strains, which seems directly linked with a gain in virulence in the guinea pig model, and an increase in resistance to hypoxia and oxidative stress. Apart from providing a long over-due explanation for the more than 50 years-old enigma on South Indian strains, our results demonstrate that the loss of the TbD1 region resulted in an important fitness gain for the L2/L3/L4 Mtb strains during host-pathogen interaction, with potential major consequences for shaping the global dominance and distribution of ΔTbD1 strains in TB epidemics. Despite comparable in vitro growth kinetics in different standard media (Middlebrook 7H9

and Dubos) under aerobic incubation conditions (Supplementary Fig. 4), the higher virulence of WT and mutant ΔTbD1 strains in aerosol-infected guinea pigs also emphasize the accuracy of the guinea pig infection model for mimicking mycobacterial virulence and severity in human disease. It is well known that guinea pigs develop hypoxic granulomatous lesions in their lungs that resemble to some extent the pathology in humans[56,57]. In these lesions, hypoxia[58] plays an important role for controlling replication of tubercle bacilli, and represents the main signal for inducing the gene transcription program regulating the shift from a replicative state to a metabolically dormant state, which ultimately results in latency[59].

This hypothesis is also supported by the results obtained from the C3HeB/FeJ mouse infection model, which is characterized by hypoxic granulomas at later stages of the infection[37–39], similar to the guinea pig infection model. ΔTbD1 Mtb strains generated a higher bacterial load than TbD1-intact strains in this model at later timepoints, whereas their virulence in the standard mouse infection models, in which granulomatous lesions do not display caseous necrosis and hypoxia, was usually comparable (Fig. 2d–f, Supplementary Fig. 7 and references[3,34]). These data are in agreement with results from the Wayne model of in vitro oxygen depletion, and suggest a link between the presence of an intact TbD1 locus and susceptibility to hypoxia of Mtb.

Our observation that TbD1-intact and ΔTbD1 WT and mutant strains showed differences in their survival after exposure to ROI and in the Wayne model of progressive oxygen depletion, but not to RNI is intriguing. Initial data on the expression profile of the *katG* gene (encoding a catalase/peroxidase responsible for $H_2O_2$ detoxification) in WT TbD1-intact Tb36 and 79112 strains and in the TbD1-deleted H37Rv revealed no significant difference among the strains, neither at basal conditions, nor at an early time point (20 min) after exposure to 10 mM $H_2O_2$ (Supplementary Fig. 14). This observation suggests that increased sensitivity of TbD1-intact strains to $H_2O_2$ is not dependent on a defect of a KatG-mediated response, a finding which is also consistent with former reports on South Indian strains, where no association between catalase activity, sensitivity to $H_2O_2$ or virulence in guinea pigs was found[31]. Moreover, we also observed comparable expression profiles for genes involved in resistance to reactive oxygen species (*ahpC* and *ahpD*) or in global transcription regulation during oxidative stress (*sigJ*), in TbD1-intact and ΔTbD1 strains, both in non-exposed and $H_2O_2$-exposed cultures. From the literature it is known that the transcriptional regulatory network induced in tubercle bacilli by the progressive oxygen depletion and required for adaptation and survival in a dormant state via the two component regulatory system DosR/DosS, overlaps with that induced after exposure to RNI[60]. In our study, no differences between TbD1-intact and TbD1-deleted Mtb strains were observed following in vitro exposure to various RNI (NaNO$_2$ and SNP), suggesting that the TbD1-mediated enhanced susceptibility to hypoxia might not be linked directly to DosR/DosS regulation mechanisms. Indeed, initial characterization of *dosR, dosS* and *dosT* gene expression in TbD1-intact 79112 and Tb36 strains in the Wayne model of in vitro oxygen depletion at different time points of incubation (day 5, day 10 and day 15) revealed no significant differences relative to the gene expression levels in ΔTbD1 Mtb H37Rv strain (Supplementary Fig. 15), further suggesting that the TbD1-related sensitivity to hypoxia might not be connected with the DosR/DosS/DosT-mediated adaptive response.

As mentioned above, the TbD1 locus encompasses the *mmpS6/mmpL6* operon, whose corresponding proteins (MmpS6/MmpL6) belong to the mycobacterial MmpS-MmpL protein family[29]. Several MmpS-MmpL proteins are involved in the trans-inner-membrane transport of lipid residues, whose encoding genes are located in the

same genomic cluster of the corresponding *mmpL* transporter encoding gene[61–69]. Intriguingly, the *mmpS6/mmpL6* gene cluster is encoded in very close genomic proximity of the *plsB1* gene (Fig. 1a), which encodes for a putative glycerol-3-phosphate acyltransferase, involved in the synthesis of the phosphatidic acid, the common intermediate in the biosynthesis of both TAG and phospholipids (such as PI, CL, or the surface-exposed PE and PIMs)[70]. However, at present it is unclear whether this genomic proximity is of any relevance, as the *mmpS/L6* operon location might have been determined by the *frdABCD* operon lateral gene transfer in selected members of the Mtb-associated phylotype[71]. Indeed, the *frdABCD* locus is absent from most mycobacterial species, including from *Mycobacterium marinum* and *Mycobacterium kansasii*, which have often been used as model organisms in mycobacterial host-pathogen research and for evolutionary comparisons[72,73]. The *frdABCD* operon encodes for the fumarate reductase, a membrane-bound bifunctional enzyme required for the maintenance of the Mtb H37Rv membrane in an energized state in anaerobic conditions in vitro[74]. While no differences were observed among TbD1-intact and TbD1-deleted strains in selected lipid profiles under the conditions tested, it cannot be excluded that the TbD1-linked *mmpS6/mmpL6* gene products might be responsible for transport of other structural mycobacterial inner-membrane-associated lipids, which may contribute to the stability of membrane-bound enzymatic complexes (e.g., fumarate reductase) involved in the adaptive response to limiting oxygen. The transport of such lipids outside the inner membrane might be detrimental for the mycobacterial survival under hypoxia. The TbD1-deletion-mediated inactivation of the MmpS6/MmpL6 transport system in the common ancestor of L2/L3/L4 strains, seems to have conferred to modern Mtb strains an evolutionary advantage for the adaptation to limiting oxygen availability. In this context, the increased susceptibility of TbD1-intact strains to oxidative stress might be a secondary effect of the extracellular export of MmpS6/MmpL6-transported substrates. Our study thus opens new perspectives for designing future experimental work focusing on more detailed aspects of such transport, taking into consideration also the initial acquisition of the *frdABCD* locus by horizontal transfer to members of the Mtb-associated phylotype[71], and the presence of a point mutation in *mmpL6* of animal-adapted strains of the MTBC[2].

A previous transcriptional analysis of the Mtb reference strain H37Rv by Betts and co-workers found the truncated *mmpL6* gene (*rv1557*) up-regulated (ratio 2.16) by exposure to Triclosan at 1×MIC for 2h[75]. This molecule with antimicrobial activity inhibits the enoyl-ACP reductase InhA and is also thought to generate oxidative stress. In the same line, a recent study by Arumugam and colleagues has used a dynamic luciferase promoter assay to evaluate the induction of various *mmpL* genes of Mtb under different drug exposure conditions and thereby also identified the promoter of MmpS6/MmpL6 operon as responding to triclosan exposure, as well as to Plumbagin[76]. The authors further report that in their setting an Mtb strain belonging to lineage L1 and an L3 strain complemented with a plasmid harboring an intact *mmpS6/mmpL6* operon showed enhanced tolerance to triclosan and plumbagin relative to a WT L3 strain. The authors then conclude that clinical Mtb strains naturally expressing intact MmpS6/MmpL6 proteins show enhanced resistance to oxidative stress[76], a finding which is in stark contrast with our results. While we have also noticed a strong difference in susceptibility to oxidative stress induced by hydrogen peroxide between MmpL6-intact and MmpL6-truncated strains, we observed that L1 strains with an intact *mmpS6/mmpL6* operon were much less tolerant to oxidative stress than the *mmpl6*-truncated L2/L3/L4 strains. The reasons for these diverging findings are not known, but we would like to emphasize that our results are in very good agreement with the results generated long before any data on genomic differences between Mtb strains were known[2,32], and also

fully support the epidemic character of the globally distributed L2/L3/L4 strains. Indeed, it would be difficult to explain why obvious beneficial means of intrinsic resistance to oxidative stress would have been lost during Mtb-host co-evolution towards the most widely distributed strain lineages[29]. As large-scale oxidative and hypoxic stress network analyses were usually undertaken with reference strains from the ΔTbD1 L4 phylogenic lineage, such as Mtb H37Rv[59,77], our results also argue for including TbD1-intact Mtb L1 strains in future such work.

On a more evolution-oriented note, the question when the deletion of the TbD1 region might have occurred during the evolution of the MTBC, remains highly speculative, as the estimations for the divergence dates of the MTBC range from 70,000 to 6000 years, depending on the model used[5,78]. However, genotypic analysis of the earliest known prehistoric case of tuberculosis in Britain, dated to the middle period of the Iron Age, ~2200 years before present, suggested that the causative strain was an Mtb ΔTbD1 strain[79]. This suggests that ΔTbD1 Mtb strains are already circulating in Europe for more than 2000 years, whereby L4 strains are thought to have caused the great majority of cases during the years of extreme high prevalence of tuberculosis in Europe during the 18 and 19th century[80]. Moreover, analysis of the evolutionary history of Mtb strains in China suggested that the current tuberculosis epidemic in the country originates from historical human migration events dating back around 1000 years before present, which established strains of lineages L2, and to a lesser extent L4, in China[49]. Taken together, it seems that the loss of the TbD1 region from the ancestor of L2/L3/L4 strains had occurred already several thousands of years ago, and provided the basis for efficient spread in emerging, densely populated human settings.

In conclusion, by the construction and use of Mtb strains belonging to different lineages and their corresponding knock-in and knock-out derivative constructs, we show here that the loss of the TbD1 region is connected with the ability of ΔTbD1 strains to better cope with oxidative stress and hypoxic conditions, thereby generating an important advantage for the bacterium during host-pathogen interaction, particularly during prolonged stages of infection. These in vitro findings, which are also consistent with results from guinea pig and C3HeB/FeJ mouse infection experiments, provide an appealing hypothesis for explaining—at least in part—the global predominance of L2/L3/L4 Mtb strains, and link the early observations from the 1960s on British and South Indian Mtb strains to a global phenomenon with major impact on the shaping of the global tuberculosis epidemic.

## Methods

**Bacterial strains and culture conditions.** *Escherichia coli* DH5α and HB101 strains were used for cloning procedures in pBluescript II SK (+) (Invitrogen) vector or pJV53-zeo plasmid[81] propagation were cultured in Luria Bertani (LB) liquid or solid media. When required kanamycin (Sigma), zeocin (Invitrogen), hygromycin (Invitrogen) and ampicillin (Sigma) were added to a final concentration of 50 μg/ml (kanamycin), 25 μg/ml (zeocin), 100 μg/ml (hygromycin and ampicillin). Mtb reference strain H37Rv and Mtb clinical isolates were grown in Middlebrook 7H9 liquid medium, supplemented with 0.2% glycerol (w/v) and 10% enrichments containing BSA (Sigma) 5% (w/v), glucose (Sigma) 2% (w/v) and NaCl (Sigma) 0.85% (w/v). For growth on solid medium, Middlebrook 7H11 agar medium was used supplemented with glycerol 0.5%, and 10% oleic acid-albumin-dextrose-catalase enrichments (OADC). When needed, kanamycin, hygromycin or zeocin were added to a final concentration of 20 μg/ml, 100 μg/ml, 25 μg/ml, respectively. Plates used for CFU counting of lung homogenates contained a PANTA (Becton Dickinson) antibiotic mixture. Inspection of Middlebrook 7H11 agar plates for CFU counting of different bacterial strains in various experimental conditions was undertaken after 3 and 5 weeks of incubation.

**Genome sequencing.** Genomic DNA was extracted from cultured single bacterial colonies as described by Cole et al.[82]. For genome sequencing of Mtb strain 79112 Sanger reads from 10-kb fragment shotgun libraries were assembled with contigs obtained from Newbler assemblies of 454/Roche reads using Arachne[83]. Scaffolds

were validated using an inhouse Mekano interface developed at Genoscope. Primer walking, PCR and in vitro transposition were used for finishing purposes. The assembled consensus sequences were validated using Illumina reads. A high-quality, contiguous genome sequence of 4394 kb in 9 contigs were generated for strain 79112. Remaining gaps estimated not to exceed 2 kb. For Mtb strains Tb36 and Mtb79499 Illumina 36-bp single-end reads were assembled using Velvet[84] and generated contigs were ordered according to similarity with the reference genome of Mtb H37Rv[82]. Genome assemblies of Mtb strains 79112, 79499 and Tb36 have been deposited in the EMBL/ENA database under the study accession number PRJEB30653. Sequences will also be made accessible via the MicroScope Magnifying Genome (MaGe) web server[85], https://www.genoscope.cns.fr/agc/microscope/home/index.php.

**Construction of TbD1 deleted and complemented Mtb strains.** The Mtb 79112ΔTbD1 mutant strain was generated by using the "Recombineering" strategy[35]. Briefly, 500-bp fragments (corresponding to the 5'- and 3'-TbD1-homologous regions) and a 1473-bp fragment (corresponding to the kanamycin resistance gene) were obtained by PCR from Mtb 79112 genomic DNA and the pUC4K plasmid, respectively. Amplified fragments were digested with *SpeI/XbaI, XbaI, XbaI/NotI* and cloned into *SpeI/NotI*-digested pBluescript II SK(+). A 2476-bp linear fragment (containing the kanamycin resistance gene flanked by the 5'- and 3'-TbD1 homologous regions) was electroporated into the Mtb 79112-pJV53-zeo recombinant strain harboring the pJV53-zeo plasmid[81], and previously incubated in culture medium supplemented with 0.2% acetamide for 16 h. Transformants were incubated in Middlebrook 7H9 medium for 3 days and then selected on Middlebrook 7H11 medium supplemented with kanamycin and zeocyn. After 3–4 weeks of incubation at 37 °C, kanamycin-resistant clones were screened by PCR for TbD1 deletion. The selected Mtb 79112ΔTbD1 mutant was cultivated in absence of zeocin for the cure of the pJV53-zeo plasmid. For construction of Mtb TbD1-complemented strains, the integrative cosmid 2G12, carrying the TbD1 locus and flanking regions from Mtb Tb36, was used. For the construction of cosmid 2G12, the pYUB412 vector backbone[36] was used. The Mtb 79112ΔTbD1 and H37Rv strains were transformed with five micrograms of 2G12 cosmid, and transformants were selected on Middlebrook 7H11 medium supplemented with hygromycin. The presence of an intact TbD1 locus in hygromycin-resistant clones was verified by PCR.

The sequence of primers used for cloning and screening procedures are listed in Supplementary Table 1.

**Cell infection assays.** BMDM were obtained from 8-week-old C57BL/6 mice and differentiated into macrophages by seeding them in a 96-well plate at a density of 4 × 10⁴ cells per well, and culturing them for 7 days in RPMI medium (Euroclone) supplemented with 10% heat-inactivated fetal calf serum, 10% L-cell conditioned medium, and 2 mM L-glutamine. Cells were infected with the various strains at a multiplicity of infection (m.o.i.) of 1:1. BAL-derived guinea-pig macrophages were infected with different strains at an m.o.i of 10:1 (cells:bacteria). THP-1 (ATCC) cells were cultured in RPMI medium supplemented with 10% heat-inactivated fetal calf serum (Euroclone) and 2 mM L glutamine (Euroclone). In growth assays performed with wild-type ancestral and modern strains, THP-1 cells were differentiated/activated into macrophages by seeding them in 96-well plates at a density of 7.5 × 10⁴ cells/well, and incubating them with 50 nM PMA for 1 day. Cell infection was performed at a m.o.i of 1:10 (bacteria: cells). For the comparison of the intracellular growth profiles of the panel of TbD1-mutant and recombinant strains, THP-1 cells were differentiated/activated into macrophages by seeding them in 96-well plates at a density of 4 × 10⁴ cells/well, and incubating them with 100 nM PMA for 3 day. Cell infection was performed at a m.o.i of 1:20 (bacteria: cells) or 1:1 (bacteria: cells). A549 (ATCC) cells were cultured in DMEM medium supplemented with 10% heat-inactivated fetal calf serum (Euroclone) and 2 mM L-glutamine (Euroclone), and then seeded in 96-well plates at a density of 4 × 10⁴ cells/well for 16 h before the infection. Different bacterial strains were added at an m.o.i of 1:10 (bacteria: cells) or 1:1 (bacteria: cells). Phagocytosis was carried out for 4 h (BMDM, BAL-derived macrophages and A549 cell line) or 1.5 h (PMA-activated THP-1 cells). In all ex vivo models, the number of intracellular bacteria (CFU) was determined immediately after phagocytosis and at different time points (as indicated in the corresponding figure panels) by lysis of cell-monolayers with 0.1% Triton X-100 in PBS. Serial 10-fold dilutions of cell lysates were then plated on Middlebrook 7H11 medium for CFU counting.

**In vivo animal studies.** Outbred Hartley guinea-pigs (Charles River) (8 per group) were used for virulence. Animals were aerosol-challenged with bacterial suspensions containing 5 × 10⁶ CFU/ml (high-dose infection) or 1 × 10⁶ CFU/ml (low-dose infection) of different Mtb strains, to obtain expected inhaled dose of 500 bacilli/lungs or 100 to 200 bacilli/lungs, respectively. The CFU numbers in lungs and spleen of infected animals were determined after infection (day 1) and 8 weeks post-challenge.

For infection studies in the murine model, six/eight-week-old female C57BL/6 mice (Charles River) were infected via aerosols generated from a suspension containing 1–5 × 10⁶ CFU/ml of different Mtb strains, to obtain an inhaled dose of 50–100 CFU/lungs. Thirty days after infection, the bacterial load in lungs and spleens was determined. C3HeB/FeJ mice (Jackson Laboratories) were infected by aerosol with bacterial suspensions to obtain an inhaled dose of 10–20 CFU/lungs.

The bacterial load in target organs (lungs and spleen) was determined 4, 10, and 14 weeks after infection.

**Ethics statement**. All animal studies were performed in agreement with European and French guidelines (Directive 86/609/CEE and Decree 87–848 of 19 October 1987). The study received the approval by the Institut Pasteur Safety Committee (Protocol 11.245) and the ethical approval by local ethical committees "Comité National de Réflexion Ethique sur l'Expérimentation Animale N° 59 (CNREEA)" or "Comité d'Ethique en Experimentation Animale Institut Pasteur N° 89 (CETEA)" (CNREEA 2012-0061; CETEA 2013-0036, CETEA dap160018, CETEA dap180023).

**Histology**. Histological preparations of lungs from C3HeB/FeJ mice, at 14 weeks post infection with TbD1-intact and TbD1-deleted Mtb strains. After fixation in 10% neutral buffered formalin for 24h-48h, organs were embedded in paraffin for 4-μm sectioning according to standard procedures and stained in in hematoxylin and eosin (HE). Stained slides were evaluated with axio scan.Z1 and Zen software (Zeiss).

**In vitro susceptibility tests**. Sensitivity tests to reactive oxygen or nitrogen intermediates (ROI and RNI) or acidic pH were performed on mycobacterial cultures in exponential growth phase. Cultures were diluted in fresh Middlebrook 7H9 broth to an OD600 = 0.1, corresponding to $5 \times 10^6$ CFU/ml, and incubated in presence of: (i) 5 mM and/or 10 mM $H_2O_2$; (ii) 10 mM sodium nitroprusside (SNP); (iii) 1 mM and 10 mM sodium nitrite ($NaNO_2$). Sensitivity test to $NaNO_2$ were performed on 10-fold diluted cultures, in fresh Middlebrook 7H9 medium previously acidified to pH 5.5 by addition of 2N HCl. At different incubation times the number of viable bacteria in treated and non-treated control cultures was determined. For each strain, the susceptibility was expressed as percentage of survival, calculated as the ratio between the CFU recovered from cultures exposed to stress over those obtained in unexposed cultures × 100. Susceptibility tests to cumene hydroperoxide and plumbagine were performed by a zone diffusion assay[86]. Aliquots of 100 μl of cultures in exponential growth phase, corresponding to $3 \times 10^6$ CFU, were spread on the surface of Middlebrook 7H11 medium. A 6.5-mm diameter paper disk, saturated with 10 μl of solution of 350 mM cumene hydroperoxide diluted in DMSO, or 5 mM plumbagine, was placed in the center of the agar plate. Control assays were performed using a paper disk saturated with 10 μl of DMSO. Plates were incubated at 37 °C for 3 weeks. The bactericidal effect of cumene hydroperoxide and plumbagine was determined by measuring the diameter of the halo of growth inhibition.

Growth/survival assays in hypoxic conditions were performed according the Wayne dormancy culture model[46]. Mtb strains were grown in Dubos medium without glycerol, to exponential phase ($OD_{600}$ = 0.4). Aliquots of 200 μl of each culture were re-inoculated into 20 ml of fresh Dubos medium without glycerol in sealed 30 ml tubes. These growth conditions were set-up in order to obtain an air-head space of 10 ml, corresponding to a head space ratio (HSR) of 0.5. Tubes were thus incubated at 37 °C in slow agitation (40 rpm/min). Immediately after inoculum (day 0) and at different incubation times (day 2, 5, 10, 25, and 40), the number of viable bacteria was determined.

**RNA extraction and RT-PCR quantification assays**. RNA was obtained from bacterial cultures using the TRIZOL reagent (Thermo Fisher) following the manufacturer's instructions with minor modifications[87]. Bacteria were broken in 1 ml of Trizol (Invitrogen) in presence of zirconia beads (0.1 mm diameter) in a MM300 apparatus (Qiagen) (30 s at maximum speed). RNA was obtained by extraction with 0.2 vol of chloroform and precipitation for 1 h at – 80 °C with 0.1 vol of 3 M sodium acetate and 0.45 vol of isopropanol. Removal of contaminating DNA was performed using DNA*free* kit (Ambion), according to the manufacturer's instructions. The absence of DNA contamination in RNA preparations was confirmed by PCR analysis using 16S rRNA specific primers. The RNA was thus reverse transcribed into cDNA (RevertAid First Strand cDNA Synthesis Kit, Thermo Fisher). Briefly, 1 μg of RNA was incubated for 5 min at 65 °C with 1 μl of N6 random primers in 12 μl final volume. A mix reaction containing 4 μl 5 × RT Buffer, 2 μl dNTP mix, 1 μl RNAase inhibitor, 1 μl reverse transcriptase, was added to each sample. The reaction was incubated at 45 °C for 1 h, followed by incubation at 72 °C for 5 min. To perform qRT-PCR expression analyses, 2-fold dilutions of *M. tuberculosis* genomic DNA were used as PCR templates for construction of standard curve and determination of the efficiency of each primer combination. Each standard dilution and each sample were tested in duplicate/triplicate and negative controls were included in each experiment. Each amplification was performed in a CFX96 Touch Real-Time PCR Detection System instrument (Bio-Rad). Quantification reactions (20 μl) contained 24 ng of cDNA template and 0.5 μM specific primers in SsoAdvanced Universal SYBR Green Supermix (Bio-Rad), 1 μl Mix containing primers. The reaction was subjected to denaturation at 95 °C for 2 min followed by 40 cycles of denaturation at 95 °C for 45 s, annealing/elongation at 66 °C (*katG* and *sigA*) or 62 °C (*sigA*, *ahpC*, *ahpD*, *sigJ*, *dosR*, *dosS*, *dosT*) for 1 min. Fluorescent data were specified for collection at the end of the annealing/elongation step in each cycle. For each gene, a standard curve of amplification was performed using known amount of DNA from H37Rv. Based on the CT values obtained, a linear regression line was plotted using a custom-made software supplied with the Bio-Rad CFX96 Touch Real-Time PCR Detection System instrument, and the

resulting equation is used to calculate the efficiency of amplification. To normalize the results between samples, the *sigA* gene and Mtb H37Rv were used as reference gene and reference strain, respectively. The expression level of each gene was reported as ratio of expression (fold induction) compared to the expression level in different strains, using following calculation: ratio of induction = Efficiency of amplification (gene of interest)$^{- \Delta\Delta CT}$, where $\Delta CT$ = CT(gene of interest) − CT (*sigA*) in each strain tested, and $\Delta\Delta CT$ = ΔCT(strain tested) − ΔCT(H37Rv).

Sequence of primers used in qRT-PCR assays are listed in Supplementary Table 1.

**TLC and MALDI lipid fingerprinting**. Heat-killed mycobacteria were first washed 3 times with PBS. The pellets were then submitted to $CHCl_3$/MeOH 1:2 (v/v) extraction for 12 h at room temperature followed by one $CHCl_3$/MeOH 1:1 (v/v) extraction and one $CHCl_3$/MeOH 2:1 (v/v) extraction for 3 h at room temperature. Pools of extracts were concentrated and evaporated to dryness. Total lipids extracted were normalized to the dried weight of lipid extract. The dried lipid extracts were resuspended in $CHCl_3$ at a final concentration of 20 mg/mL. 5 μL, equivalent to 100 μg, were loaded on TLC. TLC were run in three different solvent systems $CHCl_3$/MeOH 9:1 (v/v); $CHCl_3$/MeOH/$H_2O$ 60:25:4 (v/v/v) and $CHCl_3$/ MeOH/$H_2O$ 30:8:1 (v/v/v). Then, TLCs were sprayed with a solution of 5% phosphomolybdic acid in 100% ethanol and heated at 200 °C for 2 min in order to reveal the lipids.

In order to analyze the cell wall mycolic acids (MAMEs) the delipidated bacterial pellets were mixed with 2 ml 15% TBAH (tetrabutyl ammonium hydroxide) solution and incubated at 100 °C for 5 h. After cooling down to room temperature for 1 h, samples were incubated with 2 ml of chloroform and 100 μl of iodomethane for 1 h for methylation of the released mycolic acids. The organic lower layer after centrifugation at $2000 \times g$ at 4 °C for 10 min were collected and dried under a stream of N2. Dried MAMES were resuspended by 3 ml diethyl ether and mixed thoroughly. All supernatants were collected after centrifugation at $2,000 \times g$ at 4 °C for 10 min, transfer into a new glass tube and dried under a stream of $N_2$.

Pellets were resuspended in 50 μl of $CHCl_3$ and 5 ml of MeOH and left in the 4 °C for overnight in order to precipitate MAMES. MAMES were collected after centrifugation at $2,000 \times g$ at 4 °C for 10 min and dried under a stream of $N_2$. The purified MAMES were suspended in $CHCl_3$ at a final concentration of 10 mg/ml and analyzed by MALDI mass spectrometry in the positive ion mode as described below.

For lipid fingerprint, 5 mL of heat-inactivated mycobacterial culture was resuspended in 500 μl of distilled water, washed three times with double distilled water and resuspended in 100 μl of double distilled water. 0.4 μl of the mycobacterial solution was loaded onto the target and immediately overlaid with 0.8 μL of a 2, 5-dihydroxybenzoic acid (DHB) matrix used at a final concentration of 10 mg/mL in chloroform/methanol ($CHCl_3$/MeOH/TFA) 90:10:0.1 v/v/v. Mycobacterial solution and matrix were mixed directly on the target by pipetting and the mix was dried gently under a stream of air. MALDI-TOF MS analysis was performed on a4800 Proteomics Analyzer (Applied Biosystems) using the reflectron mode. Samples were analyzed by operating at 20 kV in the positive and negative ion mode using an extraction delay time set at 20 ns.

**Statistical analysis**. Data analysis was first performed by using the Bartlett's test and the Levene's test. According to results obtained, analysis of variance (one-way Anova) followed by Bonferroni post hoc test (in vitro survival assays and mouse infection studies) or *F*-test for heteroscedasticity (guinea-pig virulence studies) were performed. Based on results obtained d with the *F*-test, the univariate analysis of variance test followed by Sidak post hoc tests was performed. A *P*-value ≤0.05 was considered significant.

**Reporting summary**. Further information on research design is available in the Nature Research Reporting Summary linked to this article.

## Data availability
Genome assemblies of Mtb 79112, 79499 and Tb36 have been deposited in the EMBL/ENA database under the study accession number PRJEB30653.

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

## Acknowledgements
We are grateful to Professor Denis Mitchison, passed away in July 2018 aged 98, who initially enabled this study by kindly providing us with some of his original South Indian Mtb strains. We also thank Cristina Gutierrez for sharing selected Mtb clinical isolates, Dick van Soolingen for providing Mtb Tb36 strain, William R. Jacobs for the gift of vector pYUB412 and Graham F. Hatfull for providing vector pJV53. We thank Fabien Le Chevalier, Laleh Majlessi, Alexander Pym, Paul Wheeler, Stephen Gordon for help and stimulating discussions, Magali Tichit, Sabine Maurin and Johan Bedel from the histology service of the Institut Pasteur for their technical support, Alessandro Massolo for helpful advice in statistical analyses, and Jean Jaubert for help with the setup of the C3HeB/FeJ model. This work was supported in part by the French National Research Council ANR (ANR-10-LABX-62-IBEID, ANR-16-CE35-0009, ANR-16-CE15-0003), the Fondation pour la Recherche Médicale (DEQ20130326471 and SPF20160936136) and the European Union's Horizon 2020 Research and Innovation Program (643381).

## Author contributions
D.B., and R.B. designed research. D.B., W.F., F.S., M.D.L., D.S., A.P., M.Z., D.H., G.L.M., and R.B. performed experiments. M.O., V.K., S.M., V.B., C.M., L.M., C.B. performed genome sequencing studies. D.B., W.F., F.S., M.D.L., A.P., M.O., G.L.M., A.T., and R.B. analyzed data. The paper was written by D.B. and R.B. with contributions from all authors.

## Competing interests
The authors declare no competing interests.
