## [Peer Review File · Nature Communications]

Reviewers' comments:

Reviewer #1 (Remarks to the Author):

In this manuscript Bottai and colleagues studied the role of the TbD1 region of *M. tuberculosis* in the outcome of infection. The TbD1 region is present in ancestral strains found in restricted geographical areas (in East Africa and South East Asia) but is deleted in globally spread modern *M. tuberculosis* strains.

Previous reports have shown that the TbD1 intact strains were less virulent in infection in guinea pigs and were more susceptible to oxygen stress. In this manuscript, the authors confirm those reports and also generate a panel of TbD1 knocked out and knocked in strains. These recombinant strains reproduce the biological behaviour of the unmodified (non-recombinant) strains. Thus, the "attenuated" phenotype of the ancestral strains can be ascribed to the presence of the TbD1 region. While aims (understanding *M. tuberculosis* evolution and adaptation) of the research are interesting and important and the paper is well written, in my view, the study provides rather minor advances over prior knowledge. Some of the observations performed in the study also remained unexplained. The molecular mechanisms behind the TbD1-mediated reduced virulence were not explored, and while speculated in the discussion the role of the molecules encoded in the region remain unexplained. Whether there is/ was any adaptive advantage of the TbD1 region for the ancient *M. tuberculosis* is likely but unclear. There are quite many questions to be answered so is difficult to make a priority for me but some of the following aspects could have been addressed.

Specific comments.

1. A kinetic analysis of infection on the guinea pig should have been added, including earlier time points after infection. A survival curve could have been added as well. Similarly, only one time point after infection was shown for mice. Thus the result only reflects a single time point for both infection models and might not mirror the infection development.
2. Differences in pathology could also have been measured. The distribution of bacteria in granulomas or disseminated inflammation elsewhere in the lung could have been determined. As well the distribution of bacteria in solid (usually non hypoxic) and caseous granulomas could have been indicated.
3. Given results on hypoxic conditions in axenic cultures whether the different strains and the TbD1 intact or deleted develop latency in vivo in the necrotic or solid granulomas could have been studied by PCR or expression arrays. Expression of genes associated to dormancy or reactivation could have been compared here.
4. Just to help in clarifying the discussion, while developing hypoxic granulomas (which are morphologically more similar but still rather different to human granulomas), the GPs are very

susceptible to *M. tuberculosis* infection and do not develop a latent infection, but dormant bacteria have been shown.

5. The macrophage data could be included as a main figure if properly completed. Still the reasons by which human cell line and the murine macrophages differ in the control of TbD1 intact or deleted bacteria could have been better explored. For example, better comparisons could have been done using primary human macrophages rather than a cell line. These studies could also have been done with the recombinant strains. Difference in uptake and growth clearly measured at different time points after infection. The activation of these cells with cytokines (i.e. IFN- γ) could have added also another aspect of the outcome of infection and the differential effect of antioxidants in the model should have been studied. Hypoxia mimetics could also have been used in this studies.

6. Whether either mmp6S and/ or mmpL6 account for the attenuated virulence of the intact TbD1 strains could have been tested by constructing the proper recombinant strains.

7. The expression and function of *M. tuberculosis* enzymes and molecules involved in mycobacterial oxidative homeostasis and ROS detoxification in normal conditions or under stress could have been evaluated. Also the accumulation of ROS and in *M. tuberculosis* should have been assessed.

8. As a suggestion for formatting, I think the manuscript would have read better if data is ordered as follows: 1. Description of the genetic manipulations 2. Biological results in both strains and recombinants. The second part could be ordered in 1: axenic cultures, 2: in vitro macrophage infections and 3rd and last virulence in animal models.

Reviewer #2 (Remarks to the Author):

Bottai D et al. Nature Communications NCOMMS-19-12018-T

An important manuscript, further supporting the case that 'modern' *M. tuberculosis* strains, responsible for the global spread of TB epidemics, are lacking the *Mtb*-specific deletion 1 region (known as TbD1), in contrast to strains of *Mtb* with intact TbD1 regions, still in existence but confined to limited geographical regions, and presumable less virulent (some information could be included on the extent of disease caused by these S. India/E.Africa strains).

A meticulous review of that earlier/background, extensively published, information creates a review-type manuscript, unwieldy, protracted, especially in the Discussion, and tends to diminish the importance of the new information (also relegation of much of the new important data to the Supplement is not helpful; those supplementary tables, S2 and S3 are not needed).

A strengths of the new information lies in the construction of a panel of recombinant TbD1 KO and KI strains; however, these are not well characterized genetically or bacteriologically (it is not clear that these were previously characterized).

Results arising from infection of guinea pigs with the constructed, laboratory and human strains from S. India and E. African, of which the status of the TbD1 region is known, are convincing and well documented (Fig S1 or Fig. S4 could well be included in the main manuscript). Most impressive is the data with Mtb of the 79112 background demonstrating that deletion of the TbD1 locus resulted in up to 2-log increase in CFU over the wild-type strains with an intact locus. Also convincing are data demonstrating increased sensitivity of the TbD1-intact strains to ROS.

Initially disconcerting was the evidence of comparable CFU values in mice infected with the TbD1-intact versus the TbD1 deleted strains. However, the reason, well explained in the Discussion, suggests that part of this discussion should be presented in the Results as the possible reason for the lack of concordance between the two animal TB models.

A major weakness is in the effort to attribute the virulence differences to changes in lipid profile. The rationale for the approach is very weak. Why concentrate on PIMs and SL analysis only (Fig S7)? There are so many other more prominent 'virulence' lipids; any thought on a proteomic analysis? This worthy approach should be prefaced with a description of the entire spectrum of lipids and proteins encoded by the TbD1 locus. Note that the MmpS-MmpL protein family may be involved in the export of a range of Mtb secondary products not just PIMs and SLs.

Reviewer #3 (Remarks to the Author):

Bottai and co-authors present a substantial body of work exploring the function of the TbD1 locus in *Mycobacterium tuberculosis*. This locus is found in so-called 'ancient' lineages of *M. tuberculosis*, but its function has been cryptic since it was first described over ~17 years ago. Here the authors link the TbD1 locus to increased sensitivity to ROS and hypoxia, showing that deletion of the locus in 'modern' *M. tuberculosis* lineages may contribute to the relative greater success of these lineages in transmission and disease. Furthermore, they show that the reduced virulence and increased sensitivity to ROS of 'south Indian' strains of *M. tuberculosis*, first described by Denis Mitchison over 50 years ago, is due to the presence of an intact TbD1 locus. As such the manuscript both solves a question that has persisted for many decades, and also provides functional insight to a key evolutionary branch point that distinguishes 'ancient' from the more successful 'modern' *M. tuberculosis* lineages.

Major comments:

1. The specificity of the response is very interesting, with 'TbD1+' wild type strains and recombinants having increased sensitivity to ROS but not ROI. While the authors obviously invested considerable time in trying to elucidate the mechanistic basis for the TbD1+ phenotype, what do the authors speculate may be the reason for this selective sensitivity to ROS? Did the authors check the expression of, for example, katG in their mutant vs wild type to see if this gene played any role? Or other genes known to be implicated in ROS but not RNI responses?
2. The animal work is convincing, and the use of the guinea pig model nicely shows how phenotypes that are only assayed in mice may miss crucial differences. However, could the authors comment on why high and low dose models were used? Is this a matter of sensitivity?
3. As well as looking at export of lipids, which seems to have been uninformative, did the authors try to explore other potential export functions of the mmpSL6 system? Were there any differences in colony morphology between the TbD1+ or TbD1- variants?

Minor comments

1. Line 94: The introduction of the south Indian strains from Dr D Mitchison is a bit abrupt here. A short sentence or two describing the significance of these strains is warranted in the introduction, and the sequencing data can then be moved to Results. The phrase 'opened completely new perspectives' seems too strong, and instead could be better phrased as "opened new opportunities for comparative studies etc"
2. Line 131 and 134: Log reductions are discussed here, but as the experiments are not comparing isogenic strains (e.g. wild type and mutant) I think it would be better to just say "lower levels", so "~2 log lower CFUs were recovered from the lungs of M. tuberculosis...".
3. Line 391: "strains an evolutionary advantage.." rather than evolutive
4. The discussion is quite long and could be improved by shortening to ensure that the conclusions are clear. For example, the section from lines 435-451 discussing the various functions of other RD loci is quite long and not totally relevant.

Point to point responses for manuscript Nature Communications NCOMMS-19-12018-T

Please find below our detailed responses to the different points (in blue) and the associated changes in the manuscript (in green), which are also highlighted in the submitted revised marked manuscript

Reviewer #1 (Remarks to the Author):

In this manuscript Bottai and colleagues studied the role of the TbD1 region of *M. tuberculosis* in the outcome of infection. The TbD1 region is present in ancestral strains found in restricted geographical areas (in East Africa and South East Asia) but is deleted in globally spread modern *M. tuberculosis* strains.

Previous reports have shown that the TbD1 intact strains were less virulent in infection in guinea pigs and were more susceptible to oxygen stress. In this manuscript, the authors confirm those reports and also generate a panel of TbD1 knocked out and knocked in strains. These recombinant strains reproduce the biological behaviour of the unmodified (non-recombinant) strains. Thus, the "attenuated" phenotype of the ancestral strains can be ascribed to the presence of the TbD1 region. While aims (understanding *M. tuberculosis* evolution and adaptation) of the research are interesting and important and the paper is well written, in my view, the study provides rather minor advances over prior knowledge. Some of the observations performed in the study also remained unexplained. The molecular mechanisms behind the TbD1-mediated reduced virulence were not explored, and while speculated in the discussion the role of the molecules encoded in the region remain unexplained.

Whether there is/ was any adaptive advantage of the TbD1 region for the ancient *M. tuberculosis* is likely but unclear. There are quite many questions to be answered so is difficult to make a priority for me but some of the following aspects could have been addressed.

We thank the reviewer for the time and efforts taken to read and comment on our manuscript. We are pleased that the aims (understanding *M. tuberculosis* evolution and adaptation) of our research are considered by the reviewer as interesting and important.

Concerning the comment that our study provides rather minor advances over prior knowledge, we agree that the first part of our work reports similar trends as the previous work performed in the 1960s by the eminent mycobacteria specialist Denis Mitchison. However, we have now specified in the revised manuscript that this first part was mainly done to validate our model systems.

The concerned sentence reads: line 132-136:

“Given the reduced virulence observed in the 1960s for the *M. tuberculosis* 79112 strain in guinea pigs^{31,33}, we first sought to validate our guinea pig aerosol infection model by including some of these previously used *M. tuberculosis* strains into our infection experiments. We thus generated and analyzed the virulence profiles of TbD1-intact strains (79112, 79500 and Tb36) (Fig. 1b) and compared them with those of Δ TbD1 79499 clinical and H37Rv reference strains (Fig. 1b).”

Furthermore, we would like to emphasize that our work then goes far beyond previous observations, due to the demonstration of an up-to-now unknown link between the observed increased susceptibility to H₂O₂/reduced virulence in guinea pigs and the presence of the TbD1 region in ancestral *M. tuberculosis* strains. As the work by Mitchison and coworkers was done in the 1960s, long before any reliable genetic screens and molecular strain definitions were available, the observed phenomena were associated to strains from a geographical region (i.e. Southern India), without any further information on possible genetic causes. Thus, the observations reported about Indian strains always contained a large variation of phenotypes due to the apparent presence of both TbD1+ and TbD1- strains in India, while the strains from Britain showed a much

reduced spectrum of variation, as they are essentially all TbD1-deleted strains of the “modern L4” lineages (as we know today). This discrepancy is now resolved by our results.

By the construction and characterization of defined genetic mutants, and the use of clinical isolates from different *M. tuberculosis* lineages, we demonstrate in our study that the loss of susceptibility to oxidative stress and the gain in virulence in the guinea pig model is clearly linked to the loss of the TbD1 region. We consider this association as a relevant conceptual advance over previous work, as it also changes the concept of virulence in *M. tuberculosis* strains.

While we often started our comparisons in the past from “modern” TbD1-deleted strains that are most prevalent in our collections and databases, such as H37Rv, Erdmann, CDC1551, or Beijing strains, it becomes clear from the current work that this group of “modern” TbD1-deleted strains is a subgroup of *Mtb* strains, which has apparently undergone a selection process based on increased in-vivo fitness, and became dominant in many parts of the world.

Similarly, the great majority of large data screens (e.g. for essential genes, on genes involved in hypoxia etc) have been conducted with L4 reference strains, and thus it is likely that we have missed out on some relevant factors which play a role in these processes, as we show here.

We therefore should not consider that L1 strains are less fit/virulent than L2/L3/L4 strains, but rather that L2/L3/L4 strains have become more virulent than other members of the *M. tuberculosis* complex during evolution, and this notion is also reflected in the title of our manuscript.

Our data shown in this manuscript, suggest that the greater evolutionary success of L2/L3/L4 strains was influenced by the loss of the TbD1 region, which represents a genetically clearly defined evolutionary bottleneck, and the starting point for the evolution of globally distributed TbD1-deleted *Mtb* strains. This hypothesis is based on the fact that production of reactive oxygen species and/or the development of hypoxic granulomas are well-described defense mechanisms of host’s immune response to counter *M. tuberculosis* infection. Thus, becoming more resistant to these defenses during *Mtb*-host co-evolution is likely to have given the TbD1-deleted bacteria an advantage for being selected as the most dominant and transmittable strains by the host.

We do agree that this evolutionary scenario is somehow speculative, but also wanted to point out that evolutionary scenarios, in general, often remain speculative unless paleomicrobiological evidence is available, which to the best of our knowledge is not the case here.

1. A kinetic analysis of infection on the guinea pig should have been added, including earlier time points after infection. A survival curve could have been added as well. Similarly, only one time point after infection was shown for mice. Thus the result only reflects a single time point for both infection models and might not mirror the infection development.

We agree with the reviewer that a kinetic infection analysis in guinea pigs (GP) would have likely provided more informative data on earlier time points of infection, but we also would like to point out that the infection of GPs with *Mtb* in a BSL3 facility is a very resource- and space-intensive experiment for which we have had restricted availabilities to increase the numbers of GPs in our BSL3 animal facility. The presented results of several rounds of experiments have - in our opinion - shown that there is a clear strain-dependent difference in virulence in GPs that is linked with the presence or absence of the TbD1 region, and this was also confirmed by the use of genetic deletion and complementation constructs. Additional GP experiments on the same research subject would be now difficult to justify in view of the animal ethics principles of 3 Rs.

However, to try to comply with the reviewer’s suggestions we have undertaken additional virulence studies in the recently developed C3HeB/FeJ mouse model (Harper et al., JID, 2012). Different to standard mouse models, C3HeB/FeJ mice have been reported to develop granulomatous lesions which evolve into hypoxic granulomas at later time points, 10 -14 weeks after low-dose infection with *M. tuberculosis* (Harper et al., JID, 2012). The comparison of the in vivo growth kinetics of TbD1-intact and TbD1-deleted derivative strains in C3HeB/FeJ mice was undertaken by assessing the bacterial load in lungs and spleen of infected mice 4, 10 and 14 weeks after infection, which correspond to the acute and chronic phase of infection, respectively (Harper et al., JID, 2012).

These results are now presented in Fig. 6 and Supplementary Fig. 8a and are described in the paragraph that reads: lines 232-254:

“TbD1 deletion significantly enhances the virulence of *M. tuberculosis* recombinants in C3HeB/FeJ mice

To get deeper insights into potential factors that might have contributed to the divergent results obtained for virulence comparisons of TbD1-intact and Δ TbD1 strains in guinea pigs and C57BL/6 mice, we selected *M. tuberculosis* strains 79112, 79112 Δ TbD1, 79112 Δ TbD1-C and H37Rv, to evaluate their virulence in the C3HeB/FeJ mouse model, known for developing hypoxic necrotic lung granulomas 8 to 14 weeks after low-dose infection with *M. tuberculosis*³⁷⁻³⁹. Hence, C3HeB/FeJ were infected with different strains (10-20 CFU/lungs), and the bacterial load in selected organs was determined 4, 10 and 14 weeks after infection, which correspond to the acute and chronic phase of infection³⁸. Comparable CFU numbers were recovered from mice infected with TbD1-intact strains (79112 and 79112 Δ TbD1-C) and Δ TbD1 strains (79112 Δ TbD1 and H37Rv) at 4 weeks of infection (Fig. 6a, b and Supplementary Fig. 8a), confirming previous observations in the standard C57BL/6 mouse model. However, a significant lower bacterial load was observed for strain 79112 in the lungs and spleens of these mice in comparison with 79112 Δ TbD1 and H37Rv strains after 14 weeks of infection (Fig. 6 c, d), confirming a trend observed already when bacterial counts were determined in organs at 10 weeks post-infection (Fig. 6b). At these long-term time points post infection, mice infected with strain 79112 Δ TbD1-C showed an overall reduced level in bacterial load in lungs and spleens at 10 and 14 weeks post infection. Inspection of histopathological sections of the lungs showed that all of the tested strains induced granuloma formation in this model, known for some heterogeneity among individual mice (Supplementary Fig. 9)”

However, in order to keep the numbers of main figures within the limits, we have moved some of the previously shown mouse data to the supplementary material, and have combined previous Figure 5 and previous Supplementary Figure 6, these results are now shown in Supplementary Figure 7 of the revised manuscript.

2. Differences in pathology could also have been measured. The distribution of bacteria in granulomas or disseminated inflammation elsewhere in the lung could have been determined. As well the distribution of bacteria in solid (usually non hypoxic) and caseous granulomas could have been indicated.

We agree with the reviewer that information on pathology can provide additional information, although in most known cases of Mtb infection, the bacterial load is proportionally linked to the pathology caused. That is why we had included in the original manuscript examples of images of organs, on which the extent of granulomas and the spreading out to other organs (e.g. spleen) can be seen. It should also be mentioned that most of the lungs of the guinea pigs contained a lot of blood influx, due to the killing by progressive CO₂ exposure (one recommended way of euthanasia of guinea pigs in our ethical protocols), and thus histological examination was not undertaken. The different pathologies caused by TbD1 intact and TbD1 deleted strains in guinea pig organs, as well as the influx of blood can be seen in Supplementary Fig. 1 and Supplementary Fig. 6 of the revised manuscript. In addition, images of mouse lungs are depicted in Supplementary Figure 7, on which a relatively even distribution of granuloma in these lungs caused by different strains are visible.

However, to further take into consideration the reviewer’s suggestion, for the newly added model of C3HeB/FeJ mice, histopathology images from lungs are now shown in Supplementary Fig. 9 and briefly mentioned in paragraph in lines 251-254, which reads as follows:

“Inspection of histopathological sections of the lungs showed that all of the tested strains induced granuloma formation in this model, known for some heterogeneity among individual mice (Supplementary Fig. 9)”

3. Given results on hypoxic conditions in axenic cultures whether the different strains and the TbD1 intact or deleted develop latency *in vivo* in the necrotic or solid granulomas could have been studied by PCR or expression arrays. Expression of genes associated to dormancy or reactivation could have been compared here.

We thank the reviewer for this suggestion. In response to this point, an initial comparison of the expression profiles of *dosR*, *dosS* and *dosT* genes (key players in the adaptive response to hypoxia and hypoxia-induced dormancy in *M. tuberculosis*) was performed in wild-type TbD1-intact strains (79112 and Tb36) and the TbD1-deleted H37Rv reference strain in the Wayne model of *in vitro* hypoxic cultures. These initial results suggest that the lower survival of TbD1-intact strains under hypoxia do not seem to be related to potential defects in the expression of the *dosR/dosS/dosT* genes. These data are included in the revised version of the manuscript in Figure Supplementary Figure 15, and these results have also been mentioned in a paragraph in the discussion that reads as follows (lines 444-449):

“Indeed, initial characterization of *dosR*, *dosS* and *dosT* gene expression in TbD1-intact 79112 and Tb36 strains in the Wayne model of *in vitro* oxygen depletion at different time points of incubation (day 5, day 10 and day 15) revealed no significant differences relative to the gene expression levels in TbD1-deleted H37Rv strain (Supplementary Fig. 15), further suggesting that the TbD1-related sensitivity to hypoxia might not be connected with the DosR/DosS/DosT-mediated adaptive response.”

Finally, we would like to announce that the analysis of global gene expression profiles of TbD1 intact or TbD1-deleted strains in hypoxic conditions in axenic cultures and/or in *in vivo* granulomas will be investigated in a future study specifically dedicated at investigating the molecular mechanism responsible to the TbD1-associated sensitivity to hypoxia.

4. Just to help in clarifying the discussion, while developing hypoxic granulomas (which are morphologically more similar but still rather different to human granulomas), the GPs are very susceptible to *M. tuberculosis* infection and do not develop a latent infection, but dormant bacteria have been shown.

We agree with the reviewer and have changed the wording accordingly in the abstract and the discussion section.

The sentences now read: lines 41-45:

“By constructing and characterizing a panel of recombinant TbD1-knock-in and knock-out strains, here we show that deletion of the TbD1 region confers to Δ TbD1 *M. tuberculosis* strains an enhanced virulence in the sensitive guinea-pig and C3HeB/FeJ mouse infection models, which both mirror to some extent the development of hypoxic granulomatous lesions in human disease progression.”

and lines 411-412:

“It is well known that guinea pigs develop hypoxic granulomatous lesions in their lungs that resemble to some extent the pathology in humans^{55,56}.”

5. The macrophage data could be included as a main figure if properly completed. Still the reasons by which human cell line and the murine macrophages differ in the control of TbD1 intact or deleted bacteria could have been better explored. For example, better comparisons could have been done using primary human macrophages rather than a cell line. These studies could also have been done with the recombinant strains. Difference in uptake and growth clearly measured at different time points after infection. The activation of these cells with cytokines (i.e. IFN-) could have added also another aspect of the outcome of infection and the differential effect of antioxidants in the model should have been studied. Hypoxia mimetics could also have been used in this studies.

In response to these suggestions, we would like to point out that use of primary human macrophages has advantages but also disadvantages as donor-derived variation may also play a role. In addition, the setup and adaptation of this system would need more time than that which is foreseen for the revision of the manuscript.

Following the suggestions, we have moved the description of cell culture and infection procedures to the Methods section of the manuscript, but we would prefer to retain the macrophage infection results performed with wild-type TbD1-intact (79112 and Tb36) or TbD1-deleted (79499 and H37Rv) strains in the supplementary material section.

However, in response to the suggestion of the reviewer, we have undertaken additional infection experiments, where uptake and intracellular growth kinetics of TbD1-intact and TbD1-deleted mutant strains were investigated in two different cellular models, namely the PMA-activated THP-1 human macrophage and in A549 Type II pulmonary epithelial cell lines. These results are now shown in Figure 4. While results on uptake into THP-1 or A549 cells showed no differences between the tested TbD1-intact and TbD1-deleted wild-type and mutant strains, the TbD1-deleted strains (79112 Δ TbD1 and H37Rv) displayed increased intracellular growth as compared to the wild-type or recombinant TbD1-intact strains in these ex vivo models. These results are now described in lines 198-209 of the revised manuscript and read as follows:

“We next evaluated the potential effect of the presence or absence of the TbD1 region on intracellular growth kinetics of the different mutant strains in two different cellular models, namely the PMA-activated THP-1 human macrophage and in A549 Type II pulmonary epithelial cell lines. No differences were detected among TbD1-intact and Δ TbD1 strains in the percentage of uptake by THP-1 or A549 cells, at the multiplicity of infection tested (m.o.i. 20:1 and 1:1 cell:bacteria for THP-1, and m.o.i. 10:1 and 1:1 cell:bacteria for A549 cells) (Fig 4a, d). However, increased intracellular growth was observed over a 6-day period for the TbD1-deleted mutant 79112 Δ TbD1 as compared to the corresponding WT and complemented strains (79112 and 79112 Δ TbD1-C), both in THP-1 (m.o.i 20:1) and A549 (m.o.i 10:1) cells (Fig. 4b, c, e, f). Similarly, the complementation of H37Rv with a functional TbD1 locus resulted in a significant reduction of the intracellular growth ability, in both cellular models analyzed (Fig. 4b, c, e, f).”

6. Whether either mmp6S and/ or mmpL6 account for the attenuated virulence of the intact TbD1 strains could have been tested by constructing the proper recombinant strains.

In response to this suggestion, we would like to point out that the genomic organization of the various mmpS/mmpL loci (including the one of mmpS6-mmpL6 in TbD1 intact strains) suggest that these two genes are organized as an operon and work together. To make this clearer, we have included a new reference (Melly and Purdy, 2019) on the organization of the mmpS/mmpL transporters in the manuscript that was recently published and nicely reviews the organization of the mmpS/mmpL proteins, and which supports this view.

7. The expression and function of M. tuberculosis enzymes and molecules involved in mycobacterial oxidative homeostasis and ROS detoxification in normal conditions or under stress could have been evaluated. Also the accumulation of ROS and in M. tuberculosis should have been assessed.

In response to this suggestion, in association with a request from reviewer 3, an initial analysis of the expression profiles of selected genes involved in ROS detoxification (e.g. *katG*) or in regulation of global transcription during the oxidative stress (e.g. *sigJ*) was performed in wild-type TbD1-intact Tb36 and 79112 strains as well as in the H37Rv reference strain, both in basal conditions and at early-time after exposure to H₂O₂. The data obtained are shown in supplementary Fig. 14 and are mentioned in the revised version of the manuscript (please see below also the response to reviewer 3). The paragraph reads as follows (lines 425-437):

“Our observation that TbD1-intact and Δ TbD1 WT and mutant strains showed differences in their survival after exposure to ROI and in the Wayne model of progressive oxygen depletion, but not to RNI

is intriguing. Initial data on the expression profile of the *katG* gene (encoding a catalase/oxidase responsible for H₂O₂ detoxification) in wild-type TbD1-intact Tb36 and 79112 strains and in the TbD1-deleted H37Rv revealed no significant difference among the strains, neither at basal conditions, nor at an early time point (20 min) after exposure to 10 mM H₂O₂ (Supplementary Fig. 14). This observation suggests that increased sensitivity of TbD1-intact strains to H₂O₂ is not dependent on a defect of a KatG-mediated response, a finding which is also consistent with former reports on “South Indian strains”, where no association between catalase activity, sensitivity to H₂O₂ and virulence in guinea pigs was found³¹. Moreover, we also observed comparable expression profiles for genes involved in resistance to reactive oxygen species (*ahpC* and *ahpD*) or in global transcription regulation during oxidative stress (*sigJ*), in TbD1-intact and ΔTbD1 strains, both in non-exposed and H₂O₂-exposed cultures.”

Moreover, we would like to point out that it would be interesting to test many more parameters, but we suggest that this as a subject of a study apart, like this is for example shown in a paper by Nambi et al. (CHM, 2015), who have deciphered the Oxidative Stress Network of Mycobacterium tuberculosis H37Rv or Galagan et al., (Nature 2014) for hypoxia. As both these analyses were done with Mtb H37Rv (a TbD1-deleted strain of L4), we have now included a suggestion into the discussion that future work should also include L1 strains in such global analyses. The paragraph reads as follows: lines 504-509:

“Indeed, it would be difficult to explain why obvious beneficial means of intrinsic resistance to oxidative stress would have been lost during *M. tuberculosis*-host co-evolution towards most widely distributed strain lineages²⁹. As large-scale oxidative and hypoxic stress network analyses were usually undertaken with reference strains from the TbD1-deleted L4 phylogenetic lineage, such as *M. tuberculosis* H37Rv^{76,58}, our results also argue for including TbD1-intact *M. tuberculosis* L1 strains in future such work.”

8. As a suggestion for formatting, I think the manuscript would have read better if data is ordered as follows: 1. Description of the genetic manipulations 2. Biological results in both strains and recombinants. The second part could be ordered in 1: axenic cultures, 2: in vitro macrophage infections and 3rd and last virulence in animal models.

Concerning the suggestion to change the order of the data presented in the manuscript, we would like to point out that the order of sections was chosen in relation to the connection that we have established between guinea pig studies from the 1960s and a specific genomic region (the TbD1 region), discovered much later.

This is also the reason why we have structured the paper as such, starting with validating our guinea pig model by testing the 79112 strain in comparison with other TbD1-intact and TbD1-deleted strains. Our scientific reasoning was based on this link and thus has guided the choices for the construction of the knock-in and knock-out mutants and the combinations of strains/mutants that we have then used in the different phenotypic assays and animal experiments. For this reason, we would prefer, if possible to keep this part of the manuscript structure in the original form.

In case there is a strong feeling about this suggestion, we may also change the order as suggested, but in this case the history of reasoning, which has led to this manuscript would have been lost.

Reviewer #2 (Remarks to the Author):

Bottai D et al. Nature Communications NCOMMS-19-12018-T

An important manuscript, further supporting the case that 'modern' *M. tuberculosis* strains, responsible for the global spread of TB epidemics, are lacking the Mtb-specific deletion 1 region (known as TbD1), in contrast to strains of Mtb with intact TbD1 regions, still in existence but

confined to limited geographical regions, and presumably less virulent (some information could be included on the extent of disease caused by these S. India/E.Africa strains).

We thank the reviewer for the time and efforts taken to comment on our manuscript and for the consideration of our manuscript as an important manuscript. We agree with the reviewer it would be helpful to add more clinical information on L1 strains. However, as tuberculosis is a very complex disease, it is also the host which might play an important role for outbreak of disease. By looking through the literature one can notice that L1 strains are able to cause pulmonary TB as well as extrapulmonary TB in people. However, there seems to be a geographic association, and a tendency that L1 strains are involved in endemic infections but do not represent globally transmitted epidemic strains. Following the suggestion of the reviewer, we have now added some references on this subject and a sentence that reads as follows: (lines 72-73):

“The L1 strains can be subdivided in numerous sublineages^{19,20} and can cause pulmonary TB as well as extrapulmonary TB in susceptible populations^{21,22}.”

A meticulous review of that earlier/background, extensively published, information creates a review-type manuscript, unwieldy, protracted, especially in the Discussion, and tends to diminish the importance of the new information (also relegation of much of the new important data to the Supplement is not helpful; those supplementary tables, S2 and S3 are not needed).

We thank the reviewer for this comment and point of view, and have streamlined the discussion section of the manuscript, by deleting selected sentences, mainly in the previous paragraph that described the potential phenotypes of regions of differences (RD regions), which was also a request from reviewer 3 (point 4 of minor comments- please see below).

Concerning supplementary tables S2 and S3, we thought that this information might be useful for the reader interested in more details of genes involved in the hypoxic conditions, however, if this information is contributing to the impression of a review-like structure of our research paper, in agreement with the reviewer, we have deleted Supplementary tables S2 and S3.

A strengths of the new information lies in the construction of a panel of recombinant TbD1 KO and KI strains; however, these are not well characterized genetically or bacteriologically (it is not clear that these were previously characterized).

We thank the reviewer for the point of view that the construction and use of the KO and KI mutants contribute important new information for the manuscript. It is true that the mutants were constructed within this study and have not been published before. As regards the genetic characterization of the mutants, we wanted to refer to Fig 4 of the original manuscript (Fig. 3 of the revised manuscript), where the KO mutants are described and PCR data that indicate the successful genetic deletion of the TbD1 region, are shown. We have now added more information on the integrative cosmid that was used to obtain the H37Rv::TbD1 strain and this information is now shown in Supplementary Fig. 3.

Concerning the bacteriological characterization of the mutants, we would like to refer to previous Supplementary Fig. S9 of the original manuscript (Supplementary Fig. 4 of the revised manuscript), where the in vitro growth curves of WT and mutant strains in liquid growth medium are shown and no obvious differences in their growth characteristics compared to wildtype were observed.

However, to comply with the reviewer's request to add further information, we have now also added the pictures of colonies, which was also a request of reviewer 3 (major comment No. 3).

These images were added to Supplementary Fig. 5.

We have also added some text to the results section, to explain and emphasize these data. The paragraph reads (lines 191-197) as follows:

“In an assessment of their in vitro growth abilities, the various WT, mutant and complemented strains were grown in different liquid growth media (e.g. Middlebrook 7H9 and Dubos media) at different

temperatures (37°C and 42°C), where they displayed comparable growth kinetics (Supplementary Fig. 4). In contrast, TbD1-intact *M. tuberculosis* 79112 and Tb36 strains displayed smaller colony morphotypes on solid media compared to the Δ TbD1 H37Rv strain (Supplementary Fig. 5).”

Results arising from infection of guinea pigs with the constructed, laboratory and human strains from S. India and E. African, of which the status of the TbD1 region is known, are convincing and well documented (Fig S1 or Fig. S4 could well be included in the main manuscript. Most impressive is the data with Mtb of the 79112 background demonstrating that deletion of the TbD1 locus resulted in up to 2-log increase in CFU over the wild-type strains with an intact locus. Also convincing are data demonstrating increased sensitivity of the TbD1-intact strains to ROS.

We thank this reviewer for this appreciation of our data, and agree to move the data from Supplementary Fig. 1 and Supplementary Fig. 4 to the main figures of the manuscript. These data have been added to Fig. 2 in the revised manuscript.

Initially disconcerting was the evidence of comparable CFU values in mice infected with the TbD1-intact versus the TbD1 deleted strains. However, the reason, well explained in the Discussion, suggests that part of this discussion should be presented in the Results as the possible reason for the lack of concordance between the two animal TB models. ...

We agree that this part should be better discussed in the results section (please see below). Moreover, given the addition of CFU data from the C3HeB/FeJ mouse model, (please see comments to suggestions of reviewer 1 above), to avoid an oversaturation of the manuscript with figures showing mouse CFU data, we have moved the data from the standard mouse model to the Supplementary Material, where these data now are shown in Figure 5 and in Supplementary Figure 7 of the revised manuscript.

A major weakness is in the effort to attribute the virulence differences to changes in lipid profile. The rationale for the approach is very weak. Why concentrate on PIMs and SL analysis only (Fig S7)? There are so many other more prominent 'virulence' lipids; any thought on a proteomic analysis? This worthy approach should be prefaced with a description of the entire spectrum of lipids and proteins encoded by the TbD1 locus. Note that the MmpS-MmpL protein family may be involved in the export of a range of Mtb secondary products not just PIMs and SLs.

We agree with the reviewer that the analysis of lipid profiles from different strains should be extended. Thus, we have tried to comply with the reviewer's request by growing up cultures of the Mtb79112 WT, TbD1-KO, and TbD1-KO-complemented strains, and the Mtb H37Rv control for three weeks under aerobic and hypoxic conditions, followed by extracting total lipid preparations from the obtained mycobacterial cell pellets. These lipid preparations were subjected to TLC analyses using different solvents for separation of selected mycobacterial lipids. We have added several of these TLCs in the revised manuscript into supplementary figure 10. As described there, no clear, strain specific differences were found so far. We have also undertaken a mass spectrometry analysis of the MAMES of the different strains, but again have not found specific differences between strains. These data are also shown in a supplementary Figure 12. Finally we have also added the spectra for cultures grown under hypoxic conditions for the previous experiment, for which only the data from aerobic growth were shown in previous Supplementary figure 7. The revised Supplementary figure carries number Supplementary Figure 11. Thus, in total, 3 supplementary figures show the results for the lipid analysis that we have undertaken with selected TbD1-intact and TbD1-deleted WT and mutant strains. We also have adapted the paragraph in the manuscript accordingly. The paragraph reads as follows: (Lines 256-263):

“Search for TbD1-associated lipid substrates

Given the reported implication of several MmpS/MmpL systems in mycobacterial lipid transport^{29,40}, we evaluated whether the TbD1-encoded MmpS6/MmpL6 proteins might be involved in lipid transport, and subjected a set of WT and mutant *M. tuberculosis* strains to selected lipidomics assays,

for which the strains were cultured under aerobic or anaerobic conditions. Using different solvent conditions and thin layer chromatography (TLC), as well as MALDI-ToF mass spectrometry (MS) in the negative ion mode, lipid preparations were examined.”

We will also continue the analysis of the lipid profiles in selected WT and mutant strains as we really would be interested to identify lipids which might be involved in the process, but assume that such a study corresponds to the workload of a whole new study, that requires much larger lipid quantities and more specialized analysis capacities than we have currently at our disposal for this revision of the manuscript. We feel that even without this information on potential lipid contribution, the data that we present in this manuscript fully supports our evolutionary perspectives and conclusions.

However, following the requests of the reviewer we have analyzed profiles of lipids that might be in relation to the proteins encoded in the vicinity of the TbD1 locus. As mentioned in the discussion already, one of the neighboring genes of *mmpL6* (Rv1557) is gene *plsP1* (Rv1551), encoding for a putative glycerol-3-phosphate acyltransferase, which is thought to be involved in the synthesis of the phosphatidic acid, the common intermediate in the biosynthesis of both TAG and phospholipids. Thus, we have analyzed the above mentioned lipid preparations of the Mtb79112 WT, TbD1-KO, and TbD1-KO-complemented strains, and the Mtb H37Rv control strains under conditions, which allow different phospholipids to be separated. As mentioned above, this analysis has revealed no obvious differences in relevant phospholipid profiles. However, by genomic analyses of very closely related mycobacterial species we have found that the *mmpS6/mmpL6* region seems to be linked to a genomic island associated to the *frdABCD* fumarate reductase locus, which was apparently transferred during the evolution of *M. tuberculosis* to the recently described Mtb associated phylotype. As such, it is not clear whether the close genomic localization with gene *plsP1* (Rv1551), has any relevance or not.

We have this now indicated in the revised manuscript, in a paragraph which reads (lines 459-465):

“However, at present it is unclear whether this genomic proximity is of any relevance, as the *mmpS/L6* operon is thought to have been acquired together with the *frdABCD* operon by lateral gene transfer by members of the *M. tuberculosis*-associated phylotype⁷⁰. Indeed, the *frdABCD – mmpS6/mmpL6* locus is absent from most mycobacterial species, including from *Mycobacterium marinum* and *Mycobacterium kansasii*, which have often been used as model organisms in mycobacterial host-pathogen research and for evolutionary comparisons^{71,72}.”

In conclusion, in response to the reviewer's criticisms, we have prepared and analyzed new lipid preparations which have for the moment not revealed potential new candidates for lipid transport by *MmpL6*. We have also observed some differences in PDIM fractions between L1 and L4 strains, which however might be due to previously described differences in gene *rv2962c*-between L1 and L4 strains (Krishnan *et al.*, 2011). For the moment we have considered the observation as too preliminary to be included into this manuscript. Further work is definitely needed before potential substrates for lipid transport by the TbD1 region can be proposed or not. We have therefore added the sentence: (lines 277-280):

“Hence, our initial screening of selected phospholipid profiles and MAMEs did not identify potential TbD1-associated lipid factors and suggests that dedicated fine structure analyses and/or analyses under different growth conditions will be necessary to gain deeper insights into the issue.”

Reviewer #3 (Remarks to the Author):

Bottai and co-authors present a substantial body of work exploring the function of the TbD1 locus in *Mycobacterium tuberculosis*. This locus is found in so-called 'ancient' lineages of *M. tuberculosis*, but its function has been cryptic since it was first described over ~17 years ago. Here the authors link the TbD1 locus to increased sensitivity to ROS and hypoxia, showing that deletion of the locus in 'modern' *M. tuberculosis* lineages may contribute to the relative greater success of

these lineages in transmission and disease. Furthermore, they show that the reduced virulence and increased sensitivity to ROS of 'south Indian' strains of *M. tuberculosis*, first described by Denis Mitchison over 50 years ago, is due to the presence of an intact TbD1 locus. As such the manuscript both solves a question that has persisted for many decades, and also provides functional insight to a key evolutionary branch point that distinguishes 'ancient' from the more successful 'modern' *M. tuberculosis* lineages.

Major comments:

1. The specificity of the response is very interesting, with 'TbD1+' wild type strains and recombinants having increased sensitivity to ROS but not ROI. While the authors obviously invested considerable time in trying to elucidate the mechanistic basis for the TbD1+ phenotype, what do the authors speculate may be the reason for this selective sensitivity to ROS? Did the authors check the expression of, for example, *katG* in their mutant vs wild type to see if this gene played any role? Or other genes known to be implicated in ROS but not RNI responses?

We thank the reviewer for this question. To answer it, we have first undertaken RT-qPCR analyses for *katG* (responsible for H₂O₂ detoxification) and selected genes involved in resistance to ROS (*ahpC* and *ahpD*) or in global transcription regulation during oxidative stress (*sigJ*), both in basal conditions (not exposed cultures) and at early time point (20 min) after exposure to 10 mM H₂O₂. Comparable *katG* expression profiles were observed between wild-type TbD1-intact strains (Tb26 and 79112) and the TbD1-deleted H37Rv strain, both in basal conditions and in H₂O₂-exposed cultures. These observations suggest that increased sensitivity of TbD1+ strains to H₂O₂ is not dependent from a defect in *katG*-mediated response, and to are consistent with those reported in older studies on "South Indian strains" from D. Mitchinson, describing no correlation between the susceptibility to H₂O₂ of attenuated Indian strains and their catalase activity (Ref. 31, revised manuscript.).

Moreover, comparable expression was observed in the tested strains for *ahpC*, *ahpD*, and *sigJ* genes, both in untreated and H₂O₂-treated cultures.

These observations are mentioned in the revised version of the manuscript, in a paragraph that reads (lines 425-434):

"Our observation that TbD1-intact and Δ TbD1 WT and mutant strains showed differences in their survival after exposure to ROI and in the Wayne model of progressive oxygen depletion, but not to RNI is intriguing. Initial data on the expression profile of the *katG* gene (encoding a catalase/peroxidase responsible for H₂O₂ detoxification) in wild-type TbD1-intact Tb36 and 79112 strains and in the TbD1-deleted H37Rv revealed no significant difference among the strains, neither at basal conditions, nor at an early time point (20 min) after exposure to 10 mM H₂O₂ (Supplementary Fig. 14). This observation suggests that increased sensitivity of TbD1-intact strains to H₂O₂ is not dependent on a defect of a KatG-mediated response, a finding which is also consistent with former reports on "South Indian strains", where no association between catalase activity, sensitivity to H₂O₂ and virulence in guinea pigs was found³¹."

2. The animal work is convincing, and the use of the guinea pig model nicely shows how phenotypes that are only assayed in mice may miss crucial differences. However, could the authors comment on why high and low dose models were used? Is this a matter of sensitivity?

We thank the reviewer for the comment and question. In response to this question, we have first started the mouse experiments in C57BL/6 mice with higher doses, as these mice are known to be relatively resistant to *M. tuberculosis* infection. However, as the high dose infection has resulted in infections that were very pronounced, we also tested lower doses in order to evaluate if there was any dose dependent difference seen between mutant and WT strains. Thus we have lowered the dose considerably, in follow up experiments to evaluate if the dose played a role. For the additional

C3HeB/FeJ mice we have now used a very low dose as these mice were described as highly sensible to infection with *M. tuberculosis*.

3. As well as looking at export of lipids, which seems to have been uninformative, did the authors try to explore other potential export functions of the mmpSL6 system? Were there any differences in colony morphology between the TbD1+ or TbD1- variants?

We thank the reviewer for the suggestion. Indeed there were some differences between TbD1+ and TbD1- variants in colony morphologies on solid 7H9 medium. On plates the colonies of TbD1+ WT strains and mutants were in general smaller in size, whereas the growth curves in liquid medium were comparable. A somehow similar difference in colony morphology was previously observed for a hadC knock-out mutant which showed changes in mycolic acid structures i.e. inability to synthesize methoxy forms of mycolic acids (Slama et al., Mol. Microbiol 2016), resulting in smaller colonies on agar plates. However, as mentioned above we have now tested the profiles of mycolic acids and did not detect differences between mutants and WT strains, and the virulence in mice also do not correspond to such differences.

To make this point clearer we have now depicted the images of colonies from TbD1 intact and TbD1 deleted strains in Suppl. Fig. 5, and have added following paragraph: (lines 191-197):

“In an assessment of their in vitro growth abilities, the various WT, mutant and complemented strains were grown in different liquid growth media (e.g. Middlebrook 7H9 and Dubos media) at different temperatures (37°C and 39°C), where they displayed comparable growth kinetics (Supplementary Fig. 4). In contrast, TbD1-intact *M. tuberculosis* 79112 and Tb36 strains displayed smaller colony morphotypes on solid media compared to the Δ TbD1 H37Rv strain (Supplementary Fig. 5).”

Minor comments

1. Line 94: The introduction of the south Indian strains from Dr D Mitchison is a bit abrupt here. A short sentence or two describing the significance of these strains is warranted in the introduction, and the sequencing data can then be moved to Results. The phrase 'opened completely new perspectives' seems too strong, and instead could be better phrased as "opened new opportunities for comparative studies etc"

We thank the reviewer for the suggestion, and have mentioned Dr Mitchinson's work now further upstream in the Introduction. The revised paragraph (lines 92-99) now reads:

“*M. tuberculosis* strains from South India had been thoroughly studied in the 1960s by Mitchison and coworkers, whereby important differences between these isolates from Chennai (previously Madras) and *M. tuberculosis* isolates from British tuberculosis patients were found, in terms of their virulence in the guinea pig infection model and their susceptibility to hydrogen peroxide³¹. The later independent observations that the TbD1 region was intact in most *M. tuberculosis* strains from Southern India^{32,2} prompted us to investigate whether any link between the reported attenuated phenotype in guinea pigs of tubercle bacilli from South Indian TB patients³¹ and the presence or absence of the TbD1 region might exist.”

We have also changed “opened completely new perspectives” to “opened new opportunities”, as suggested.

2. Line 131 and 134: Log reductions are discussed here, but as the experiments are not comparing isogenic strains (e.g. wild type and mutant) I think it would be better to just say "lower levels", so " ~2 log lower CFUs were recovered from the lungs of *M. tuberculosis*...".

We thank the reviewer for this suggestion, the revised sentence (line 139-142) now reads:

“A ~2-log lower CFU level was recovered from lungs of *M. tuberculosis* 79112- and Tb36-infected animal as compared to that detected in guinea pigs infected with 79499 and H37Rv *M. tuberculosis* strains (Fig. 2a, c and Supplementary Fig. 1). “

3. Line 391: "strains an evolutionary advantage.." rather than evolutive

Thank you, “evolutive” was corrected to “evolutionary “, in lane 476 of the revised manuscript.

4. The discussion is quite long and could be improved by shortening to ensure that the conclusions are clear. For example, the section from lines 435-451 discussing the various functions of other RD loci is quite long and not totally relevant.

We agree and have deleted this paragraph.

REVIEWERS' COMMENTS:

Reviewer #1 (Remarks to the Author):

The authors have properly addressed my comments to the manuscript in the reviewed version. The reviewed manuscript has improved and they have done a major effort in this. The strength is that they have properly linked the phenotype of ancient strains to the TBD1 locus. The mouse work added increases its power. However, how the genes within the locus account for their reduced virulence and enhanced susceptibility to ROI and hypoxic conditions of the L1 strains is still unravelled. The authors have tried different approaches to understand this question, and while although negative at this stage, their effort is valuable, and will be important to consider in further studies.

I have some suggestions/ comments that might improve this manuscript:

1. Clear the differences on bacterial load were observed using the C3HeB/FeJ mice. The authors have replaced data from the C57Bl/6 mice with that of C3HeB/FeJ rather than comparing it. Since there are no late time points of B6 mice, I suggest to perform this experiment or alternatively delete the information that relates to the B6 mice in the manuscript. The comparison is of interest since the Kramnik mice are deficient in a locus mediating IFN-g responses (via *Ipr1* gene).
2. Even with low infection doses used, the CFU recorded in C3HeB/FeJ seem remarkably low. Instead CFUs levels in B6 mice seem in the normal range (Suppl Fig 7). Please comment.
3. The histopathology sections must be presented in a better way. It is very difficult (too many sections, small sized) to observe differences in lesion density, or the type of lesions (encapsulated, necrotic, cellular etc) in animals infected with different strains. Just representative lesions should be shown, and a relative quantification of lesion area, number of granulomas could be indicated instead.
4. On figure 2 and 5 (panels c and f), a comparison between CFU day 1 and the day expt day is added. I understand that CFUs should have been tittered so that animals show similar CFU values at day 1, before inhalation. If so, unless you have other considerations (i.e. the CFUs at day 1 were no even), these panels can be deleted (these are not with in Figure 6 on the mouse model). The same comment is true for Supplementary Figure 7.
5. The images of gross pathology in guinea pig model in suppl fig 1 are difficult to appreciate. Differences between groups with this images are not obvious. I suggest, to remove this figure (Supplement Figure 1).
6. The revised manuscript reads better than the original version. Still, as indicated before, I think that reorganizing the manuscript will make it to read better. I suggest the following: 1. Description of the mutant construction, 2. results in both strains and recombinants. The second part could be ordered

in 1: axenic cultures, hypoxia and stress, lipid analysis 2: macrophage infections in vitro and 3rd and last virulence in animal models.

Reviewer #2 (Remarks to the Author):

Nature Communications NCOMMS-19-12018-T

The manuscript is much improved as a result of the comments of the three reviewers. The detailed responses to the various points raised by all three are among the most thorough and comprehensive I have witnessed. Consequently, revisions have involved a sizeable amount of new experimentation, and the lapse of at least 6 months since we last reviewed this manuscript.

This is a water-shed body of work whereby the functions of the TbD1 locus of *M. tuberculosis* has been thoroughly explored. The locus had been found in *Mtb* of ancient lineage but functional consequences unknown. Through the clever and skilled creation of KO and KI mutant and approaches and calling on a collection of *Mtb* isolates of modern lineage the authors have linked the TbD1 locus to increased sensitivity to ROS and hypoxia. Thus, a partial molecular explanation of the relatively greater 'virulence' (i.e. in transmission and disease induction) to today's 'modern' strains. Now, the reduced virulence and increased sensitivity of the South Indian strains of 50 years ago can be attributed to the presence of an intact TbD1 locus. That in itself could have been sufficient for a water-shed manuscript.

However, all three reviewers were looking for more evidence. This reviewer, in particular, was hoping for more information on the comparative bacteriological characteristics of the two lineages, in particular efforts to equate 'virulence' features with lipid phenotypes (arising from the concern that the authors selected only the PIMs and sulfolipids for investigation). The authors have made a sterling effort to address that concern with comprehensive TLC analysis and MS of mycolates. Unfortunately, nothing stands out but the manuscript is consequently much improved. Reviewer # 3 also addressed similar concerns wondering if there were major differences in colony morphology. Evidence is now presented that the TbD1+ variants and mutants show smaller size but similar growth profiles. Other concerns have been satisfactorily addressed and the supporting documentation enhanced considerably.

Surprisingly, reviewer No. 1 was not similarly impressed with the original version, writing: "it is difficult for me to make a priority"; "the study provides rather minor advances over prior knowledge". That reviewer set down an exceptionally challenging list of "aspects that could have been addressed" many of which would have resulted in at least a much different manuscript if not a new one (in response the authors say at one juncture "we would like to point out that it would be interesting to test more parameters, but we suggest that this is a subject of a study apart"). However, in response the authors convincingly responds to all expressed concerns. Most impressive

now is the demonstration that the loss of susceptibility to oxidative stress and the gain in virulence in the guinea pig model is clearly linked to the loss of the TbD1 region. The request that a kinetic analysis of infection in the guinea pig should have been included is answered based on the difficulty of justification in light of animal ethics principles under the 3 Rs. The request for demonstration of differences in pathological manifestations is answered by referring to the original manuscript in which images of organs were included showing the extent of granulomas and spreading out to spleen. However, histological images are now also included in the Supplement. Again, the request for the use of primary human macrophages rather than a cell line has been met by new experiments in two different cellular models, the PMA-activated YHP-1 human macrophage and the A549 Type II pulmonary cell line. The requests for the construction of appropriate recombinant strains to address the possible roles of the mmp transport proteins in virulence attenuation, the expression and function of enzymes and other modulators of oxidative homeostasis and ROS detoxification, and expression of the array of genes associated with dormancy or its reactivation, would of course have involved at least months of further experiments. However, the authors have stoutly tried to answer all of these concerns, in particular now including the results of an initial comparison of the expression profiles of dosR, dosS and dosG genes.

The rather unreasonable suggestion of a wholesale re-formatting of the manuscript was reasonably rejected by the authors based on the historical progression of the research story up to the present juncture.

Reviewer #3 (Remarks to the Author):

The authors have addressed my original comments, and provided a substantial amount of extra experimental data and clarifications in responses to the other reviewers. I found the revised version to be greatly strengthened.

We thank all three reviewers again for their time and insightful comments. Please find below detailed point to point responses (in blue colour) and added sentences in the re-revised manuscript (in green).

REVIEWERS' COMMENTS:

Reviewer #1 (Remarks to the Author):

The authors have properly addressed my comments to the manuscript in the reviewed version. The reviewed manuscript has improved and they have done a major effort in this. The strength is that they have properly linked the phenotype of ancient strains to the TBD1 locus. The mouse work added increases its power. However, how the genes within the locus account for their reduced virulence and enhanced susceptibility to ROI and hypoxic conditions of the L1 strains is still unravelled. The authors have tried different approaches to understand this question, and while although negative at this stage, their effort is valuable, and will be important to consider in further studies.

We thank the reviewer for the constructive comments and for the understanding.

I have some suggestions/ comments that might improve this manuscript:

1. Clear the differences on bacterial load were observed using the C3HeB/FeJ mice. The authors have replaced data from the C57Bl/6 mice with that of C3HeB/FeJ rather than comparing it. Since there are no late time points of B6 mice, I suggest to perform this experiment or alternatively delete the information that relates to the B6 mice in the manuscript. The comparison is of interest since the Kramnik mice are deficient in a locus mediating IFN-g responses (via *Ipr1* gene).

We agree with the reviewer that a longer term mouse infection in C57Bl/6 with different WT and mutant strains might have been helpful for direct comparison in the different host environments, but wanted to point out that in previous experiments in C57Bl/6 mice, similar *in vivo* growth characteristics/CFU counts of *Mtb* H37Rv and a TbD1-intact *Mtb* control strain were obtained at week 13 post infection. This information was reported in the supplementary data file of the article (Supply et al., 2013, Nat Genet), and argues for a similar virulence level between TbD1-intact and TbD1-deleted *M. tuberculosis* strains in C57BL/6 mice even at later time points post infection. Moreover, similar observations were made by an independent study in Balb/C mice, where *Mtb* strains of different lineages were compared to H37Rv (Krishnan et al., 2011, PLOS One), and where no significant difference in CFU counts between TbD1 -intact strains and H37Rv at 8 weeks post infection were found. We had already cited these two studies in the revised version of the manuscript, but have highlighted this information somehow stronger in the current revision. The revised sentence now reads:

“In contrast, no significant *in vivo* growth differences between TbD1-intact and Δ TbD1 *Mtb* strains were observed in the standard C57BL/6 murine model, in agreement with similar results from previous infection experiments involving *Mtb* H37Rv and TbD1-intact *Mtb* strains in Balb/C³⁴ and C57BL/6 mice³, which lasted up to 8 weeks³⁴ and 13 weeks³, respectively.”

Even with low infection doses used, the CFU recorded in C3HeB/FeJ seem remarkably low. Instead CFUs levels in B6 mice seem in the normal range (Suppl Fig 7). Please comment.

In response to this question, there is yet only little information found in the literature on the susceptibility of C3HeB/FeJ mice to various Mtb strain lineages, as the model is relatively recent. In a paper by Verma et al., PLOS Pathog. 2019, for example, C3HeB/FeJ mice are used in infection assays and the mice show lung CFU levels between 10^5 and 10^6 for certain lineage 4 strains, which is approximately also the level that we observe for the TbD1 deleted strains in our assay. It should be mentioned that in that assay the in-going CFU numbers are also higher than in our assay. However, in that study, no TbD1-intact strains were included. We argue in our paper that TbD1-intact strains apparently show lower CFU levels in this model. We have included the Verma et al., PLOS Pathog. 2019 reference now at the end of the paragraph.

3. The histopathology sections must be presented in a better way. It is very difficult (too many sections, small sized) to observe differences in lesion density, or the type of lesions (encapsulated, necrotic, cellular etc) in animals infected with different strains. Just representative lesions should be shown, and a relative quantification of lesion area, number of granulomas could be indicated instead.

In response to this question, we have now added images with increased magnification from selected granulomas to Supplementary Figure 9, which was thus split up into two panels a and b. The granulomas are now much better visible, thank you for the suggestion.

4. On figure 2 and 5 (panels c and f), a comparison between CFU day 1 and the day expt day is added. I understand that CFUs should have been tittered so that animals show similar CFU values at day 1, before inhalation. If so, unless you have other considerations (i.e. the CFUs at day 1 were no even), these panels can be deleted (these are not with in Figure 6 on the mouse model). The same comment is true for Supplementary Figure 7.

In response to this question, we have included this information in response to reviewer 2's request and thus would like to keep the information like it was shown in our previously revised manuscript, if possible.

5. The images of gross pathology in guinea pig model in suppl fig 1 are difficult to appreciate. Differences between groups with this images are not obvious. I suggest, to remove this figure (Supplement Figure 1).

In response to this suggestion, we want to refer to the editorial comment that no data should be in principle removed, and in our opinion, the information coming from the images are useful for appreciation of the level of severity of infection, and thus, if possible, we would like to keep the images in the supplement for interested readers.

6. The revised manuscript reads better than the original version. Still, as indicated before, I think that reorganizing the manuscript will make it to read better. I suggest the following: 1. Description of the mutant construction, 2. results in both strains and recombinants. The second part could be ordered in 1: axenic cultures, hypoxia and stress, lipid analysis 2: macrophage infections in vitro and 3rd and last virulence in animal models.

In response to this request, we would like to refer to our previous comments for the first revision of the manuscript and the comments of reviewer 2 (shown below). However, we agree that in the abstract, which is probably to most often accessed part of the article, the order of items could be changed and we have now revised the abstract accordingly, also taking into consideration the reduction of word count to 160. We hope that this might be an acceptable compromise.

Reviewer #2 (Remarks to the Author):

Nature Communications NCOMMS-19-12018-T

The manuscript is much improved as a result of the comments of the three reviewers. The detailed responses to the various points raised by all three are among the most thorough and comprehensive I have witnessed. Consequently, revisions have involved a sizeable amount to new experimentation, and the lapse of at least 6 months since we last reviewed this manuscript.

This is a water-shed body of work whereby the functions of the TbD1 locus of *M. tuberculosis* has been thoroughly explored. The locus had been found in *Mtb* of ancient lineage but functional consequences unknown. Through the clever and skilled creation of KO and KI mutant and approaches and calling on a collection of *Mtb* isolates of modern lineage the authors have linked the TbD1 locus to increased sensitivity to ROS and hypoxia. Thus, a partial molecular explanation of the relatively greater 'virulence' (i.e in transmission and disease induction) to to-day's 'modern' strains. Now, the reduced virulence and increased sensitivity of the South Indian strains of 50 years ago can be attributed to the presence of an intact TbD1 locus. That in itself could have been sufficient for a water-shed manuscript. However, all three reviewers were looking for more evidence. This reviewer, in particular, was hoping for more information on the comparative bacteriological characteristics of the two lineages, in particular efforts to equate 'virulence' features with lipid phenotypes (arising from the concern that the authors selected only the PIMs and sulfolipids for investigation). The authors have made a sterling effort to address that concern with comprehensive TLC analysis and MS of mycolates. Unfortunately, nothing stands out but the manuscript is consequently much improved. Reviewer # 3 also addressed similar concerns wondering if there were major differences in colony morphology. Evidence is now presented that the TbD1+ variants and mutants show smaller size but similar growth profiles. Other concerns have been satisfactorily addressed and the supporting documentation enhanced considerably. Surprisingly, reviewer No. 1 was not similarly impressed with the original version, writing: "it is difficult for me to make a priority"; "the study provides rather minor advances over prior knowledge". That reviewer set down an exceptionally challenging list of "aspects that could have been addressed" many of which would have resulted in at least a much different manuscript if not a new one (in response the authors say at one juncture " we would like to point out that it would be interesting to test more parameters, but we suggest that this is a subject of a study apart"). However, in response the authors convincingly responds to all expressed concerns. Most impressive now is the demonstration that the loss of susceptibility to oxidative stress and the gain in virulence in the guinea pig model is clearly linked to the loss of the TbD1 region. The request that a kinetic analysis of infection in the guinea pig should have been included is answered based on the difficulty of justification in light of animal ethics principles under the 3 Rs. The request for demonstration of differences in pathological manifestations is answered by referring to the original manuscript in which images of organs were included showing the extent of granulomas and spreading out to spleen. However, histological images are now also included

in the Supplement. Again, the request for the use of primary human macrophages rather than a cell line has been met by new experiments in two different cellular models, the PMA-activated YHP-1 human macrophage and the A549 Type II pulmonary cell line. The requests for the construction of appropriate recombinant strains to address the possible roles of the mmp transport proteins in virulence attenuation, the expression and function of enzymes and other modulators of oxidative homeostasis and ROS detoxification, and expression of the array of genes associated with dormancy or its reactivation, would of course have involved at least months of further experiments. However, the authors have stoutly tried to answer all of these concerns, in particular now including the results of an initial comparison of the expression profiles of dosR, dosS and dosG genes.

The rather unreasonable suggestion of a wholesale re-formatting of the manuscript was reasonably rejected by the authors based on the historical progression of the research story up to the present juncture.

Reviewer #3 (Remarks to the Author):

The authors have addressed my original comments, and provided a substantial amount of extra experimental data and clarifications in responses to the other reviewers. I found the revised version to be greatly strengthened.